# Hippo pathway-mediated YAP1/TAZ inhibition is essential for proper pancreatic endocrine specification and differentiation

Yifan Wu[1,2†‡], Kunhua Qin[1,3†§], Yi Xu[1], Shreya Rajhans[4], Truong Vo[4], Kevin M Lopez[1], Jun Liu[1], Michael H Nipper[1], Janice Deng[1], Xue Yin[1], Logan R Ramjit[1], Zhenqing Ye[5], Yu Luan[1], H Efsun Arda[4]*, Pei Wang[1]*

[1]Department of Cell Systems & Anatomy, University of Texas Health Science Center at San Antonio, San Antonio, United States; [2]Department of Obstetrics, The Second Xiangya Hospital, Central South University, Changsha, China; [3]Department of Molecular Medicine, University of Texas Health Science Center at San Antonio, San Antonio, United States; [4]Laboratory of Receptor Biology and Gene Expression, Center for Cancer Research, National Cancer Institute, NIH, Bethesda, United States; [5]Department of Population Health Sciences, University of Texas Health Science Center at San Antonio, San Antonio, United States

*For correspondence:
efsun.arda@nih.gov (HEA);
WangP3@uthscsa.edu (PW)

†These authors contributed equally to this work

Present address: ‡Department of Obstetrics, The Second Affiliated Hospital, School of Medicine, Zhejiang University, Hangzhou, China; §State Key Laboratory of Experimental Hematology, Department of Pharmacology, School of Basic Medical Sciences, Tianjin Medical University, Tianjin, China

Competing interest: The authors declare that no competing interests exist.

**Abstract** The Hippo pathway plays a central role in tissue development and homeostasis. However, the function of Hippo in pancreatic endocrine development remains obscure. Here, we generated novel conditional genetically engineered mouse models to examine the roles of Hippo pathway-mediated YAP1/TAZ inhibition in the development stages of endocrine specification and differentiation. While YAP1 protein was localized to the nuclei in bipotent progenitor cells, Neurogenin 3 expressing endocrine progenitors completely lost YAP1 expression. Using genetically engineered mouse models, we found that inactivation of YAP1 requires both an intact Hippo pathway and Neurogenin 3 protein. Gene deletion of Lats1 and 2 kinases (*Lats1&2*) in endocrine progenitor cells of developing mouse pancreas using *Neurog3*[Cre] blocked endocrine progenitor cell differentiation and specification, resulting in reduced islets size and a disorganized pancreas at birth. Loss of *Lats1&2* in Neurogenin 3 expressing cells activated YAP1/TAZ transcriptional activity and recruited macrophages to the developing pancreas. These defects were rescued by deletion of *Yap1/Wwtr1* genes, suggesting that tight regulation of YAP1/TAZ by Hippo signaling is crucial for pancreatic endocrine specification. In contrast, deletion of *Lats1&2* using β-cell-specific Ins1[CreER] resulted in a phenotypically normal pancreas, indicating that *Lats1&2* are indispensable for differentiation of endocrine progenitors but not for that of β-cells. Our results demonstrate that loss of YAP1/TAZ expression in the pancreatic endocrine compartment is not a passive consequence of endocrine specification. Rather, Hippo pathway-mediated inhibition of YAP1/TAZ in endocrine progenitors is a prerequisite for endocrine specification and differentiation.

## Editor's evaluation

This study presents the essential role that hippo pathway plays in proper pancreatic endocrine specification and differentiation. The authors demonstrate that the deletion of the Lats1 and 2 kinases (Lats1&2) in endocrine progenitor cells of developing mouse pancreas with Ngn3-Cre blocked endocrine progenitor cell differentiation and specification, resulting in reduced islets size

and disorganized pancreas at birth, but no effects were observed when deleting them in β cells. These results show for the first time that Hippo pathway-mediated YAP1/TAZ inhibition in endocrine progenitors is a prerequisite for endocrine specification and differentiation.

## Introduction

The mammalian pancreas is composed of exocrine and endocrine compartments, which both originate from embryonic endoderm. The exocrine pancreas is mainly comprised of acinar cells, which secrete various digestive enzymes, and ductal cells which transport these enzymes to the duodenum. The endocrine pancreas, which consists of α-, β-, δ-, PP-, and ε-cells, produces several hormones that are secreted into the blood with organism-wide functions. Of these functions, regulation of glucose homeostasis is predominately associated with pancreatic endocrine function. Developmental defects in the pancreas lead to devastating pathological diseases including congenital pancreas abnormalities, congenital hyperinsulinism, and neonatal diabetes (*Lu and Li, 2018*; *Kamisawa et al., 2017*; *Pohl and Uc, 2015*).

Initially discovered in *Drosophila*, the Hippo signaling pathway is best known for its roles in regulating organ size via suppressing growth and promoting apoptosis (*Pan, 2010*). In mammals, this pathway has more complicated, context-dependent functions with regard to regulating tissue homeostasis (*Yu et al., 2015*). The mammalian Hippo pathway is comprised of a kinase signaling cascade beginning with the Ste-20-like protein kinases (MST1&2), which directly phosphorylate and activate Large tumor suppressors 1&2 (LATS1&2) (*Yu et al., 2015*). Upon activation, LATS1&2, the final kinases in this cascade, directly phosphorylate the effectors of the Hippo pathway – the transcription coactivators Yes-Associated Protein 1 (YAP1) and WW-domain-containing transcription regulator 1 (WWTR1, also known as TAZ). The phosphorylation of YAP1 and TAZ leads to their cytosolic sequestration and/or proteasome-mediated degradation (*Meng et al., 2015*). In contrast, the inhibition of Hippo pathway stabilizes YAP1 and TAZ, enabling their cytosol-to-nucleus translocation, resulting in their association with specific transcription factors (e.g., TEAD) and initiation of related transcriptional programing (*Yu et al., 2015*). Prior studies have shown that compared with inactivation of upstream components of the Hippo pathway such as *Mst1&2*, specific genetic inactivation of *Lats1&2* more robustly facilitates YAP1 and TAZ transcriptional activities (*Meng et al., 2015*).

The Hippo signaling pathway has been shown to play a crucial role in pancreatic development, both in lineage specification and morphogenesis (*Wu et al., 2021*; *Panciera et al., 2016*). During early-stage pancreatic development, deletion of *Mst1/2* in pancreatic progenitor cells results in dysregulation of acinar expansion. De-differentiation of acinar cells into ductal-like cells (also known as acinar-to-ductal metaplasia, or ADM), immune cell infiltration, and pancreatic auto-digestion have also been observed following *Mst1/2* deletion. However, the overall function of the islets is not affected by the loss of *Mst1/2* (*Gao et al., 2013*; *George et al., 2012*). *Lats1&2* also serve vital functions in early pancreatic development before the secondary transition when endocrine differentiation reaches its peak (*Braitsch et al., 2019*). Endocrine specification is driven by the basic helix-loop-helix transcription factor Neurogenin 3 (Neurog3) in bipotent pancreatic progenitors at secondary transition stage (*Gradwohl et al., 2000*; *McGrath et al., 2015*). The *Neurog3* gene is a master regulator of pancreatic islet differentiation and initiates stepwise cell fate determination, delamination, and migration of differentiating endocrine cells (*Wang et al., 2010*; *Gouzi et al., 2011*; *Rukstalis and Habener, 2009*; *Duvall et al., 2022*). In vitro culture experiments found that Neurog3 directly represses the transcription of the *Yap1* (*George et al., 2015*). These observations lead to the notion that Hippo pathway-mediated YAP1 protein inhibition in Neurog3-positive endocrine progenitors is dispensable for the subsequent endocrine differentiation process, but this has not been examined with in vivo developmental models (*Wu et al., 2021*).

Here, we investigated the function of Hippo in endocrine cell development and homeostasis by inactivating *Lats1&2* in Neurog3+ endocrine progenitor cells and β-cells. We found that Hippo activity in Neurog3+ cells is required for mouse endocrine progenitor specification and differentiation but is not necessary for pancreatic β-cell function. By further introducing *Yap1/Wwtr1* deletion into *Lats1&2* null cells, we demonstrated that attenuation of YAP1/TAZ through genetic ablation rescued the observed phenotypes driven by *Lats1&2* inactivation. Our results uncovered a context-dependent

function of the Hippo signaling pathway in maintaining normal development and tissue homeostasis in the pancreas.

## Results

### Inactivation of *Lats1&2* in Neurog3+ cells led to defects in mouse postnatal growth, acinar atrophy, and ductal expansion in the pancreas

To determine the roles of *Lats1* and *Lats2* in endocrine progenitor cells and pancreas development, both genes were inactivated in the *Neurog3$^{Cre}$* transgenic mouse strain. For tracing of Cre-recombinase activity, we also crossed the *Rosa26$^{LSL-YFP}$* line. The heterozygote mice (*Neurog3$^{Cre}$Lats1$^{fl/+}$Lats2$^{fl/+}$Rosa26$^{LSL-YFP}$*) and single knockout mice (*Neurog3$^{Cre}$Lats1$^{fl/fl}$Lats2$^{fl/+}$Rosa26$^{LSL-YFP}$* or *Neurog3$^{Cre}$Lats1$^{fl/+}$Lats2$^{fl/fl}$Rosa26$^{LSL-YFP}$*) from the initial F1 and F2 generations showed no abnormal phenotype. We, therefore, bred the single knockout mice with the *Lats1&2* conditional allele line to get the *Neurog3$^{Cre}$Lats1$^{fl/fl}$Lats2$^{fl/fl}$Rosa26$^{LSL-YFP}$* mice (abbreviated as NL mice). The *Lats1* single knockout littermates were used as control (Ctrl) (*Figure 1—figure supplement 1A*). NL mice were born in Mendelian ratios with no obvious differences from the littermate control. However, starting from 1 week of age, it became obvious that the NL mice were smaller in size than the control mice. By the time of postnatal day 19 (P19), the NL mice appeared smaller and weaker than their littermates (*Figure 1—figure supplement 1B*), with a much smaller pancreas (*Figure 1—figure supplement 1C*), and suffered from hypoglycemia. The majority of NL mice died before 3 weeks of age, exhibiting rear limb weakness or paralysis. Feeding the mice with 10% glucose (*Xiao et al., 2010*) or keeping them with their parents delayed, but did not prevent their death (data not shown), indicating that hypoglycemia only partially accounted for the early death. A few mice survived beyond 3 weeks with much smaller body sizes and infertility. These growth-related phenotypes could be attributed to the expression of *Neurog3$^{Cre}$* in the central nervous system (*Song et al., 2010*).

Histological analysis with Hematoxylin and eosin stain (H&E) staining showed that the acinar cells, ductal system, and islets could be clearly identified in control pancreases (*Figure 1A*), whereas remarkably disorganized structures were seen in the center of the NL pancreases including noticeable expansion of ductal cells and fewer acinar cells (*Figure 1—figure supplement 1D*). Immunohistochemical staining of Insulin (INS) revealed much smaller islets in NL pancreases in addition to structural disorganization, evident by immunofluorescent co-staining of Insulin and Glucagon (GCG) (*Figure 1B*). We further analyzed the mRNA expression level of lineage-specific genes in the P1 pancreas using quantitative RT-PCR (qPCR). We found that the mRNA expression of acinar-specific markers, *Ptf1a*, Amylase (*Amy*), and carboxypeptidase A1 (*Cpa1*), as well as endocrine-specific markers, Chromogranin A (*ChrA*), Insulin 1 (*Ins1*), and Insulin 2 (*Ins2*) were significantly lower in NL mice when compared with control mice (Ctrl, $n = 4$; NL, $n = 4$) (*Figure 1C*). In comparison, while the mRNA expression of two ductal-specific markers, *Hnf1b* and *Sox9*, had no significant change, *Krt19* was significantly higher in NL pancreases (*Figure 1C*). To pinpoint the starting time of the observed defect in NL mice, we examined the E15.5 pancreas, when expression of *Neurog3* reaches its peak. We did not observe any obvious differences between NL and control pancreases at E15.5 by H&E staining. The expansion of central ductal structure was noted in the NL pancreas at E16.5 which became even more obvious at P1 from H&E staining and KRT19 immunohistochemistry staining (*Figure 1D*). Together, these observations suggest that genetic disruption of *Lats1&2* in Neurog3+ cells leads to acinar atrophy and ductal expansion in the postnatal pancreas, which are likely initiated in the developing embryonic pancreas.

### Inactivation of *Lats1&2* blocks endocrine specification and differentiation

In the developing pancreas, expression of *Neurog3* marks the beginning of endocrine cell differentiation and reaches its peak at E15.5. However, as described before, apparent defects in pancreatic morphology were not observed through H&E staining until E16.5. Therefore, subsequent analysis was focused on the E16.5 pancreas. We monitored Cre-recombinase activity with *Rosa26$^{LSL-YFP}$* allele. Yellow fluorescent protein (YFP) expression was used as an indicator for *Lats1&2* deletion as no viable antibodies were available for staining of LATS1&2. We also performed immunofluorescent staining of pancreatic and duodenal homeobox 1 (PDX1), an early pancreatic progenitor marker and β-cell-specific transcriptional factor. YFP+ cells formed cell clusters and were co-stained with PDX1 in control

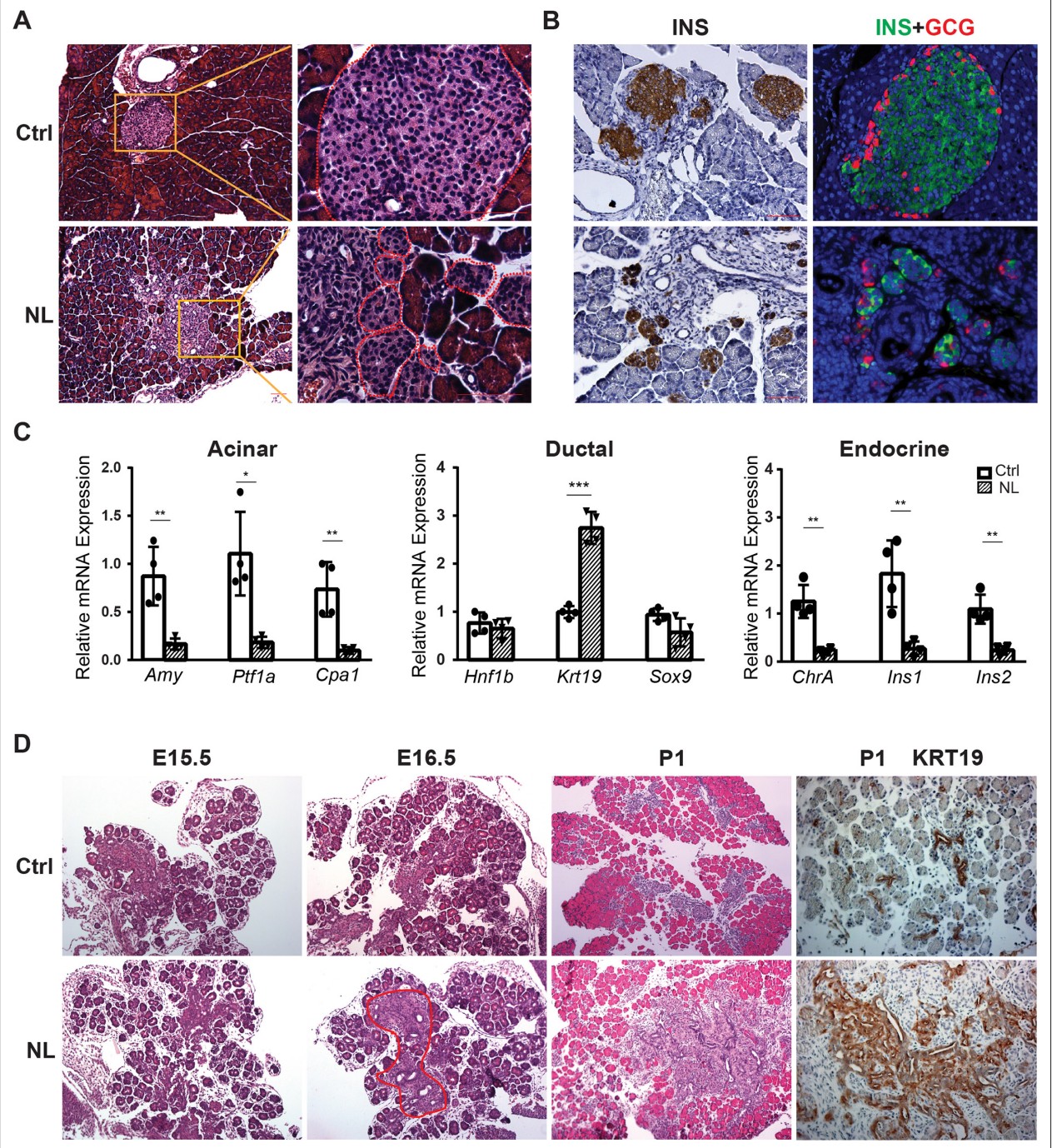

**Figure 1.** Deletion of *Lats1/2* by *Neurog3^Cre* perturbs pancreatic endocrine development. (**A**) H&E staining of pancreatic tissue from NL mice showed phenotypically abnormal pancreatic islets as compared to those in control mice. (**B**) Immunostaining of Insulin (INS) and Glucagon (GCG) showed reduced size of pancreatic islets and decreased abundance of β-cells in NL mice. (**C**) Quantitative reverse transcription real-time PCR (RT-PCR) analysis of the expression levels of acinar, ductal, and endocrine genes in neonatal (P1) control and NL pancreas (Ctrl, *n* = 4 biological replicates; NL, *n* = 4 biological replicates). *p < 0.05; **p < 0.01; ***p < 0.001. (**D**) H&E staining and immunohistochemistry staining of KRT19 of pancreatic tissue from control and NL mice at indicated stage. Structural changes in histology and a higher level of KRT19 expression in NL mice were noted at the indicated stages. Scale bar: 100 μm.

The online version of this article includes the following figure supplement(s) for figure 1:

**Figure supplement 1.** Deletion of *Lats1/2* by *Neurog3^Cre* perturbed pancreas differentiation.

pancreases (*Figure 2A*). In contrast, in NL pancreases, the majority of YFP+ cells did not form clusters, but instead formed a distinct layer next to ductal cells. In addition, there were also much fewer PDX1 and YFP double-positive cells in NL pancreases, as compared with the control (*Figure 2A*), suggesting that most endocrine progenitors in the NL pancreas do not develop into β-cells at E16.5. Presence of a few double-positive cells in NL pancreases could be due to incomplete deletion of genes. At birth, control pancreases showed co-staining of YFP and PDX1 double-positive cells in the center region of islets and YFP single-positive cells in the periphery region of islets, which is consistent with the normal structure of islet of Langerhans where β-cells localize to the center of islets while α-cells localize to the periphery. In contrast, NL pancreases had much fewer PDX1-positive cells which were found in small unorganized cell clusters, while the majority of YFP+ cells did not express PDX1 (*Figure 2—figure supplement 1*). To further delineate the defect of *Lats1/2* null endocrine progenitor cells, we performed immunostaining on the transcription factors Islet 1 (ISL1) and NK2 homeobox 2 (NKX2.2), which are expressed after Neurog3 in endocrine pancreas development (*Arda et al., 2013*). We found that ISL1 was widely expressed in YFP+ cells in control pancreases, but only expressed in a few small YFP+ cell clusters in NL pancreases, in which the majority of YFP+ cells did not express ISL1 (*Figure 2B*). A similar expression pattern was also observed for NKX2.2 (*Figure 2C*). Thus, these data indicate that loss of *Lats1&2* leads Neurog3+ endocrine progenitor cells to halt the differentiation program almost immediately and prevents them from further endocrine lineage progression.

## Loss of *Lats1&2* impairs epithelial–mesenchymal transition in endocrine progenitor cells

During the endocrine formation process, several morphogenetic events coincide with cell differentiation including epithelial-to-mesenchymal transition (EMT) and delamination of differentiating endocrine cells during pancreatic development (*Gouzi et al., 2011*; *Rukstalis and Habener, 2009*). The new model proposed by *Sharon et al., 2019* suggests that differentiating endocrine precursor cells undergo 'leaving the cord' or 'delamination' process with lower E-cadherin (CDH1) expression. How Neurog3 mediates CDH1 downregulation and facilitates 'leaving the cord' is unclear. We observed that YFP+ cells were connected to epithelial cords but expressed lower CDH1 than the cord epithelial cells in control E16.5 pancreases (*Figure 3A*). YFP+ cells in the NL pancreas were connected to the epithelial cords and formed buds or sheaths similar to YFP+ cells in the control. However, they express CDH1 at a level as high as of that in the epithelial cord (*Figure 3A*). With respect to the few escapees as previously described in PDX1 expression, the NL pancreas, too, contained a few escapees marked by lower CDH1 expression. Interestingly, we found that the YFP+ cells in NL pancreases did not stay as 'budding' or 'sheath' as development continued, and returned to single epithelial cells at P1 (*Figure 3—figure supplement 1*). These data suggest that an active Hippo pathway is required for downregulation of CDH1 expression in *Neurog3*-expressing progenitor cells, but is dispensable for 'leaving the cord' or 'delamination'.

We have observed that the differentiation of *Lats1&2* null endocrine progenitor cells was blocked by lack of expression of ISL1 and NKX2.2 transcription factors. To expand on this, we interrogated the question of whether these progenitor cells returned to the ductal cell fate. To do so, we performed immunofluorescent staining and found that, similarly to the control pancreas, there was no overlap of YFP and SOX9 expression in the NL pancreas at E16.5 or P1 (*Figure 3B*). In addition, we examined another known ductal cell marker, cytokeratin 19 (KRT19) in the NL pancreas. We observed that KRT19 was not present in YFP+ cells in control pancreases but was highly expressed in YFP+ cells of NL pancreases at E16.5 and P1 (*Figure 3C*). In fact, at E16.5, the expression of KRT19 in YFP+ cells of NL pancreases was even higher than the ductal cells, which later became similar to ductal cells at the P1 stage (*Figure 3C*). These data suggest that an early effect of *Lats1&2* deletion in Neurog3+ cells is to activate KRT19 expression, but not SOX9 expression, further indicating that KRT19 expression is not controlled by SOX9, but instead by YAP1.

## *Lats1&2* null cells recruit macrophages and induce pancreatitis-like phenotype in the developing pancreas

Following gene knockouts of *Lats1&2*, we observed that NL mice died before 3 weeks of age with a much smaller pancreas. Histological analysis showed acinar atrophy and disorganized ductal expansion in the NL pancreas, which are frequently associated with pancreatitis (*Lerch and Gorelick, 2013*).

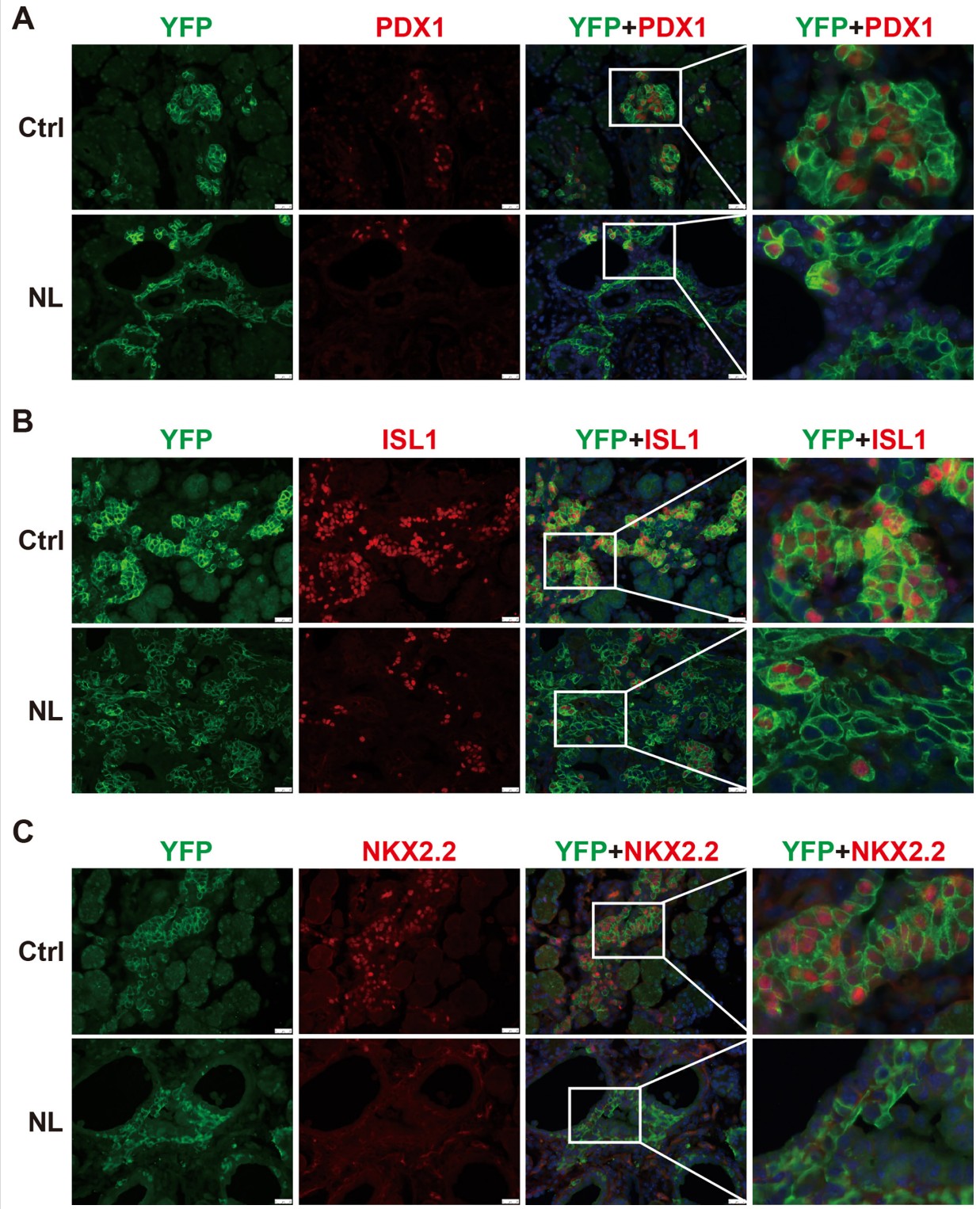

**Figure 2.** The differentiation of endocrine cells was blocked in NL pancreas at E16.5. (**A**) Immunostaining of YFP and PDX1 on pancreatic tissue demonstrated that most of *Lats1/2* null YFP+ cells in NL mice were negative for PDX1 staining. Immunostaining of YFP, Islet 1 (ISL1) (**B**), and NKX2.2 (**C**) showed that ISL1 and NKX2.2 were not expressed in the majority of YFP+ cells in NL pancreas. Scale bar: 50 μm.

The online version of this article includes the following figure supplement(s) for figure 2:

**Figure supplement 1.** The differentiation of endocrine cells was blocked in NL pancreas at P1.

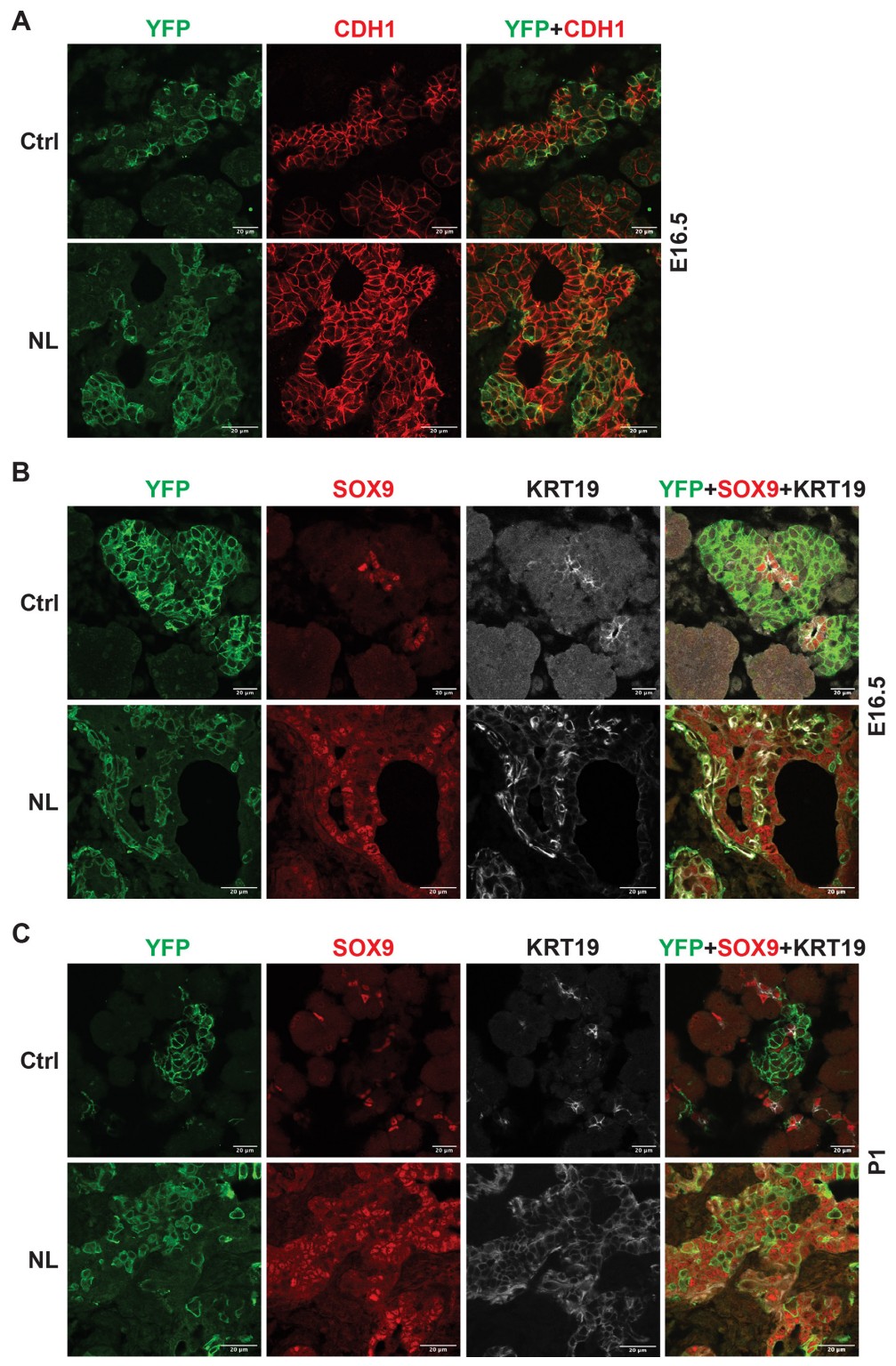

**Figure 3.** Loss of *Lats1&2* altered endocrine progenitor cell characteristics. (**A**) CDH1 protein expression in YFP+ cells was low in control pancreas and considerably higher in NL pancreas at E16.5. (**B**) In both control and NL pancreas, the YFP+ cells were negative for SOX9 staining at E16.5 and P1. (**C**) KRT19 staining was negative in YFP+ cells in control pancreas but appeared to be positive in YFP+ cells of NL pancreas at both E16.5 and P1. Scale bar: 20 μm.

The online version of this article includes the following figure supplement(s) for figure 3:

*Figure 3 continued on next page*

*Figure 3 continued*

**Figure supplement 1.** CDH1 expression in YFP+ cells was low in control pancreas and considerably higher in NL pancreas at P1.

Our previous study involving acinar-specific deletion of *Lats1&2* (using *Ptf1a^CreER^* model) has found that loss of *Lats1&2* first activates pancreatic stellate cells (PSCs) (*Liu et al., 2019*). We wondered whether a similar mechanism happened in the NL pancreas. Immunostaining showed that there was a significant increase of Vimentin-positive mesenchymal cells in the NL pancreas at E16.5 (*Figure 4A*). Those expanded mesenchymal cells were closely associated with YFP+ *Lats1&2* null cells (*Figure 4—figure supplement 1A*). These mesenchymal cells were positive for activated PSC marker alpha smooth muscle actin (ACTA2) (*Figure 4—figure supplement 1B*). Next, we determined whether there was immune cell infiltration in the NL pancreas. We observed CD45+ immune cells with large cell bodies, similar to the size of epithelial cells, in the E16.5 pancreas. We then stained tissue with macrophage marker F4/80 and found that these CD45+ cells were macrophages. The number of macrophages was increased in NL mice at E16.5, especially in the ductal area (*Figure 4B, D*), and further dramatically increased at P1 in the whole pancreas (*Figure 4C, D*). Overall, quantification analysis showed that NL pancreases contained a significantly greater density of macrophages at E16.5 and P1 in comparison to control pancreases at E16.5 and P1 in both the acinar and ductal regions of the pancreas (E16.5: Ctrl, $n = 3$; NL, $n = 3$. P1: Ctrl, $n = 4$; NL, $n = 3$) (*Figure 4D*). Together, these results indicate that loss of *Lats1&2* in Neurog3+ endocrine progenitor cells activates PSCs in the developing pancreas, and recruits immune cells, continuously.

## The expressions of YAP1/TAZ and their targets are elevated in *Lats1&2* null cells

Loss of *Lats1&2* in *Neurog3*-expressing endocrine progenitor cells blocks the endocrine lineage from further differentiation. We hypothesized that uncontrolled downstream Hippo effectors, YAP1 and TAZ, mediate this phenotype. We observed that YAP1-positive cells had increased CTNNB1 and KRT19 expression in control and NL pancreases at E16.5 (*Figure 5A*), which is consistent with our RT-qPCR experiments measuring relative mRNA expression level (*Figure 1C*). It was observed that YFP+ cells had no YAP1 expression in control pancreases at E16.5 (*Figure 5A*), suggesting YAP1 protein expression predominated in the epithelial cord region, which is consistent with previous studies (*Gao et al., 2013*; *Duvall et al., 2022*). However, YAP1 expression was much higher in NL pancreases, with many YFP+ cells co-localizing with positive YAP1 nuclei staining despite a few escaped YFP+ cells lacking YAP1 expression (*Figure 5—figure supplement 1A*). We observed a similar YAP1 staining pattern in P1 pancreases of NL mice. We further co-stained Neurog3 and YAP1 on E16.5 pancreases and found that, in the control pancreas, YAP1 was expressed in Neurog3+ cells located in the epithelial cord but was sequestered into the cytoplasm (*Figure 5B*). No YAP1 expression was detected in the cytoplasm or the nucleus in the Neurog3+ cells after they left the epithelial cord (*Figure 5B*). In comparison, much fewer Neurog3+ cells, but more YAP1+ cells, were observed in the NL pancreas, overall. In addition, YAP1 co-localized with Neurog3 in the nuclei of cells residing in the epithelial cord of NL pancreases (*Figure 5B*). Using qPCR, we then further quantified mRNA expression of the YAP1 targets that we previously identified (*Luo et al., 2016*). We found that the mRNA expressions levels of *Ankrd1*, *Cyr61*, and *TGFb1* had no significant changes, but *Ctgf*, *SPP1*, *Cxcl10*, and *Cxcl16* were significantly higher in the NL pancreas at P1 compared to control pancreas (Ctrl, $n = 4$; NL, $n = 4$) (*Figure 5C*). Immunofluorescent staining showed that SPP1 protein was expressed in ductal cells in control pancreases, but was also positive in YFP+ cells in NL pancreas at E16.5 and P1 (*Figure 5—figure supplement 1B*). Together, these data demonstrate the increased expression of YAP1 and its target genes in *Lats1&2* null cells.

## Single-nucleus RNA sequencing analysis indicated a block in the endocrine differentiation

To examine genome-wide transcriptional changes in NL mutant mice, we performed single-nucleus RNA sequencing (snRNA-seq) on E17.5 pancreata from control and NL embryos. After initial data processing and removal of low-quality cells (see Methods), we retained 1784 cells from control and

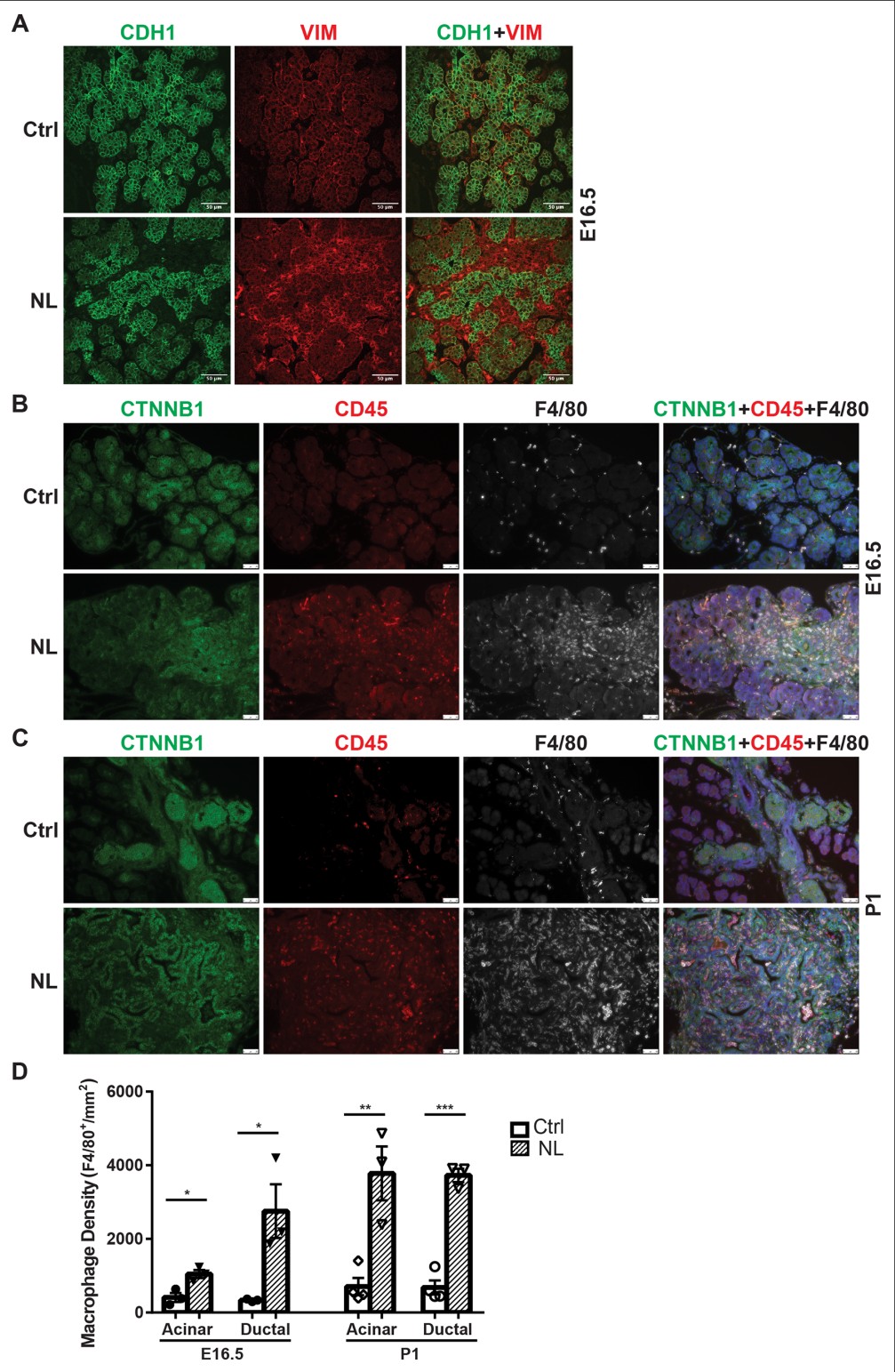

**Figure 4.** Loss of *Lats1&2* in endocrine progenitor cells was associated with an increased number of mesenchymal cells and macrophages. (**A**) Vimentin-positive staining was increased in NL pancreas at E16.5. Scale bar: 25 μm. Immunostaining showed that the number of CD45-positive immune cells and F4/80-positive macrophages was significantly increased in NL pancreas at E16.5 (**B**) and became even more apparent at P1 (**C**). (**D**) Quantification of F4/80+ macrophage showed a significant increase in NL pancreas compared to control pancreas at both E16.5 and

*Figure 4 continued on next page*

*Figure 4 continued*

P1 (E16.5: Ctrl, *n* = 3 biological replicates; NL, *n* = 3 biological replicates) (P1: Ctrl, *n* = 4 biological replicates; NL, *n* = 3 biological replicates). *p < 0.05; **p < 0.01; ***p < 0.001. Scale bar: 50 µm.

The online version of this article includes the following figure supplement(s) for figure 4:

**Figure supplement 1.** Loss of *Lats1&2* in endocrine progenitor cells was associated with increased mesenchymal cells.

2198 cells from NL mice. Unsupervised clustering and uniform manifold approximation and projection (UMAP) analysis revealed nine distinct cell types represented in both the control and NL pancreas tissue (*Figure 6A, B*, *Figure 6—figure supplement 1A*). We annotated these cell populations using hallmark cell-type-specific genes and gene expression profiles (*Figure 6—figure supplement 1B*). Two independent NL mutant embryos showed similar pattern, demonstrating the reproducibility of snRNA-seq analysis (*Figure 6—figure supplement 1C, D*). When we compared the percentage of different cell types in control versus NL mice, we observed that the ductal cell population expanded at the expense of endocrine lineage in NL mice, confirming our histology findings (*Figure 6C*). In addition, the NL pancreas displayed elevated numbers of immune cells and myofibroblasts (*Figure 6C*). Immune cell cluster is comprised of three sub-populations, including a $Il7r^+$ $Il18r1^+$ T cell cluster and two $Mrc1^+$ macrophage clusters (*Figure 6D, E*). Looking at additional macrophage marker genes showed that NL pancreas contained more $Ly86^+Lgmn^+Mertk^+$ macrophages. These results are in line with the previous findings that Yap1 activation may contribute to macrophage recruitment (*Liu et al., 2019*; *Yang et al., 2020*). Analysis of fibroblast and myofibroblast combined populations identified a total of four subclusters, including two myofibroblast clusters, a fibroblast cluster, as well as a mesothelial cluster (*Figure 6F*). Interestingly, the two myofibroblast clusters have distinctive distribution between control and NL pancreas. Specifically, the control pancreas was associated with more $Ctnna2^+Ncam1^+$ myofibroblast, whereas NL pancreas was associated with more $Hmcn2^+Dgkb^+$ myofibroblast (*Figure 6G*). Gene Set Enrichment Analysis (GSEA) showed that myofibroblast 2 of NL pancreas enriched genes for extracellular matrix organization, collagen formation and ECM–receptor interaction (*Figure 6—figure supplement 1E*). These data suggested that the loss of *Lats1&2* in endocrine progenitor cells re-shaped the tissue microenvironment.

To further characterize the differences between control and NL pancreas, we analyzed the acinar population. Three sub-clusters were identified, including a normal acinar cluster – Acinar 1, with expression of various digestive enzymes and acinar transcription factors (*Cel*, *Pnliprp1*, *Cpb1*, *Rbpjl*, and *Mecom*) (*Figure 6H, I*). This population was mainly found in control pancreas. In comparison, Acinar 2, existed in both control and NL pancreas, was found to highly express *Dlc1* but lower digestive enzymes especially *Cel*, suggesting an immature acinar state. Interestingly, Acinar 3 were mainly found in NL pancreas and express low level of digestive enzymes.

Analysis on ductal cells revealed a total of four sub-clusters, which were all associated with substantial *Onecut2*, *Sox9*, *Etv6*, and *Cdh1* expression, confirming their ductal lineage (*Figure 6J, K*). Control pancreas mainly comprised of Ductal 1, while NL pancreas has more Ductal 2 and 3, with Ductal 4 uniquely to NL pancreas (*Figure 6J*). Among these sub-clusters, Ductal 2 had the highest *Prom1* progenitor marker expression (*Figure 6K*). Interestingly, Ductal 3 showed high expression of *Notch1*, *Notch3*, and *Dclk1*, as well as endocrine cell markers *Isl1* and *Pax6*. Similar to our staining results, we found *Krt19* and *Ctnnb1* to be highly expressed in Ductal 2, 3 and 4 cells. *YAP* and *Wwtr1* were also high in these populations, however, their target gene *Ctgf* mainly expressed in Ductal 3 and 4 with highest expression in Ductal 4 of NL pancreas. Interestingly, *Sema3e*, *Csf1*, and *Pdgfb* had similar expression pattern as *Yap* target, which may indicate a potential YAP1 regulation (*Figure 6K*). In addition, differential gene expression analysis and *K*-means clustering revealed distinct gene expression patterns among the four ductal sub-populations (*Figure 6—figure supplement 1F*). Gene Ontology analysis of the Ductal 4 highly expressed genes showed enriched migration, wound healing, GTPase-mediated signaling, cell matrix adhesion, TGFβ signaling, and Hippo pathway (*Figure 6—figure supplement 1G*). These data suggest that loss of Lats1&2 expression leads Neurog3-expressing cells differentiate to a new ductal-like population.

Analysis of pancreatic cells only (acinar, ductal, and endocrine) also revealed three acinar sub-populations, four ductal sub-populations as well as endocrine α- and β-cell populations (*Figure 6L*). To study the dynamic processes of cellular development and differentiation, we performed single-cell

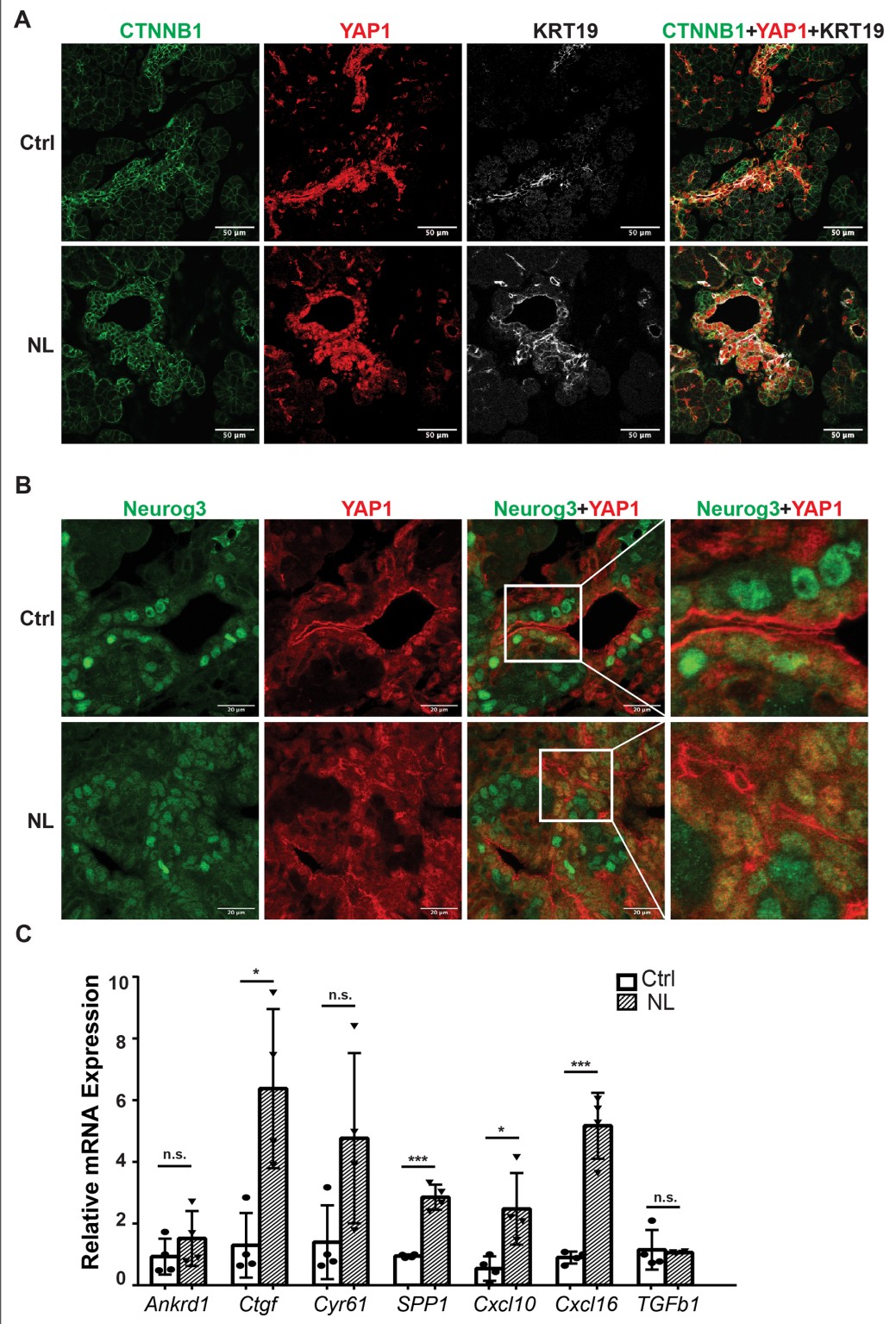

**Figure 5.** The expressions of YAP1/TAZ proteins and their targets were increased in *Lats1&2* null cells. (**A**) YAP1-positive cells showed increased CTNNB1 and KRT19 expression in both control and NL pancreases at E16.5. (**B**) YAP1 was sequestered to the cytoplasm in newly born Neurog3-positive cells located in the epithelial cord in control pancreas, while co-localization of YAP1 with Neurog3 was observed in nuclei of cord epithelial cells in NL pancreas at E16.5. (**C**) The mRNA levels of YAP1 targets *Ctgf*, *SPP1*, *Cxcl10*, and *Cxcl16* were significantly increased

*Figure 5 continued on next page*

*Figure 5 continued*

in NL pancreas at P1 (Ctrl, *n* = 4 biological replicates; NL, *n* = 4 biological replicates). *p < 0.05; ***p < 0.001. Scale bar: 50 µm (**A**) and 20 µm (**B**).

The online version of this article includes the following figure supplement(s) for figure 5:

**Figure supplement 1.** YAP1 and its targets were increased in *Lats1&2* null cells.

trajectory analysis. Because the gene expression data suggested that Ductal 2 population has progenitor cell signature in developing pancreas, we set Ductal 2 population as root and traced the possible differentiation path of ductal cells into acinar, α- and β-cells using pseudotime analysis (*Figure 6M*). The Ductal 2 path split into endocrine lineage and into Ductal 4, a population specific to NL pancreas. By checking cell cycle status, NL pancreas generally has more cells in S and G2M phases (*Figure 6—figure supplement 1H*). Specifically, Acinar 1, the major population in control pancreas, has fewer cells in S/G2M than Acinar 2 and 3, the major ones in NL pancreas (*Figure 6N*). It was also shown that a large number of proliferating ductal cells were present in NL pancreas (*Figure 6N*, *Figure 6—figure supplement 1H*). Altogether our data suggest that Lats1&2 deletion results in high YAP1 and TAZ which block further differentiation of ductal lineage, resulting in fewer endocrine cells.

## YAP1/TAZ are downstream effectors of *Lats1&2* in regulating endocrine specification and differentiation

To further demonstrate that YAP1 and TAZ are the downstream effectors of *Lats1&2* in endocrine lineage development, we performed gene knockouts of *Yap1* and *Wwtr1* in addition to *Lats1&2* in Neurog3+ endocrine progenitor cells. To this end, we intercrossed *Neurog3^Cre^Lats1^fl/fl^Lats2^fl/+^* with *Yap1^fl/fl^Wwtr1^fl/fl^* mice for several generations to obtain the following gene knockout mice models: *Neurog3^Cre^Lats1^fl/fl^Lats2^fl/+^Yap1^fl/fl^Wwtr1^fl/fl^* (named as NTY where *Lats2* is heterozygous and *Yap1/Wwtr1* are null), *Neurog3^Cre^Lats1^fl/fl^Lats2^fl/fl^Yap1^fl/+^ Wwtr1^fl/fl^* (named as NLT where *Yap1* is heterozygous but *Wwtr1* is null), and *Neurog3^Cre^Lats1^fl/fl^Lats2^fl/fl^Yap1^fl/fl^Wwtr1^fl/fl^* (named as NLTY with quadruple null *Lats1&2*, *Yap1*, and *Wwtr1*). First, we found that NTY mice were normal with no defect in the pancreas, suggesting that YAP1/TAZ are no longer needed in *Neurog3*-expressing cells for endocrine lineage development. Because NL mice showed an obvious defect in the postnatal P1 pancreas, we focused on and analyzed the P1 pancreases of NLT and NLTY mice. Histological analysis showed that, unlike NL pancreases, NLTY pancreases were similar to the control pancreas. However, NLT pancreases still showed some abnormality in the ductal network region, suggesting incomplete rescue when YAP1 expression remained (*Figure 7—figure supplement 1A*). We further analyzed endocrine cells by staining YFP, INS, and Somatostatin (SST), and observed that only a small fraction of YFP+ cells, scattered throughout the NL pancreas, were INS- and SST-positive (*Figure 7A* and *Figure 7—figure supplement 1B*). However, similar to control pancreases, most YFP+ cells in NLT and NLTY pancreases were INS- and SST-positive, suggesting that reduced levels of YAP1/TAZ rescued the endocrine defects of loss of *Lats1&2*. However, we still observed some ductal-like YFP+ cells in NLT pancreases, suggesting incomplete rescue compared to NLTY pancreases (*Figure 7A*). Furthermore, we analyzed immune cell infiltration and found that unlike NL pancreas, there were much fewer macrophages in both NLT and NLTY pancreases (*Figures 7B, 4C*). In addition, quantification analysis showed that overall macrophage density is slightly elevated in NLT and NLTY pancreases compared to control pancreases, but is not statistically significantly (Ctrl, *n* = 4; NL, *n* = 3; NLT, *n* = 3; NLTY, *n* = 3) (*Figure 7C*). Taken together, our results demonstrate that YAP1/TAZ are the downstream effectors of *Lats1&2* during endocrine differentiation, and a tight control of their activities is required for normal pancreatic development.

## *Lats1&2* are dispensable for pancreatic β-cell proliferation and function

The established role of Hippo in regulating cell proliferation prompted us to investigate whether inactivation of *Lats1&2* affects pancreatic β-cell proliferation and function. To address this, we deleted *Lats1&2* in pancreatic β-cells using the mouse-insulin-promoter-1-driven CreER (Ins1^CreER^, named as ML). We first tested whether *Lats1&2* are necessary for pancreatic β-cell function at the adult stage (*Figure 8A*). The mice were subjected to 5-day Tamoxifen (TAM) injection at 180 mg/kg/day, and pancreases were collected 4 weeks after TAM injection for histological analysis. Normal pancreatic

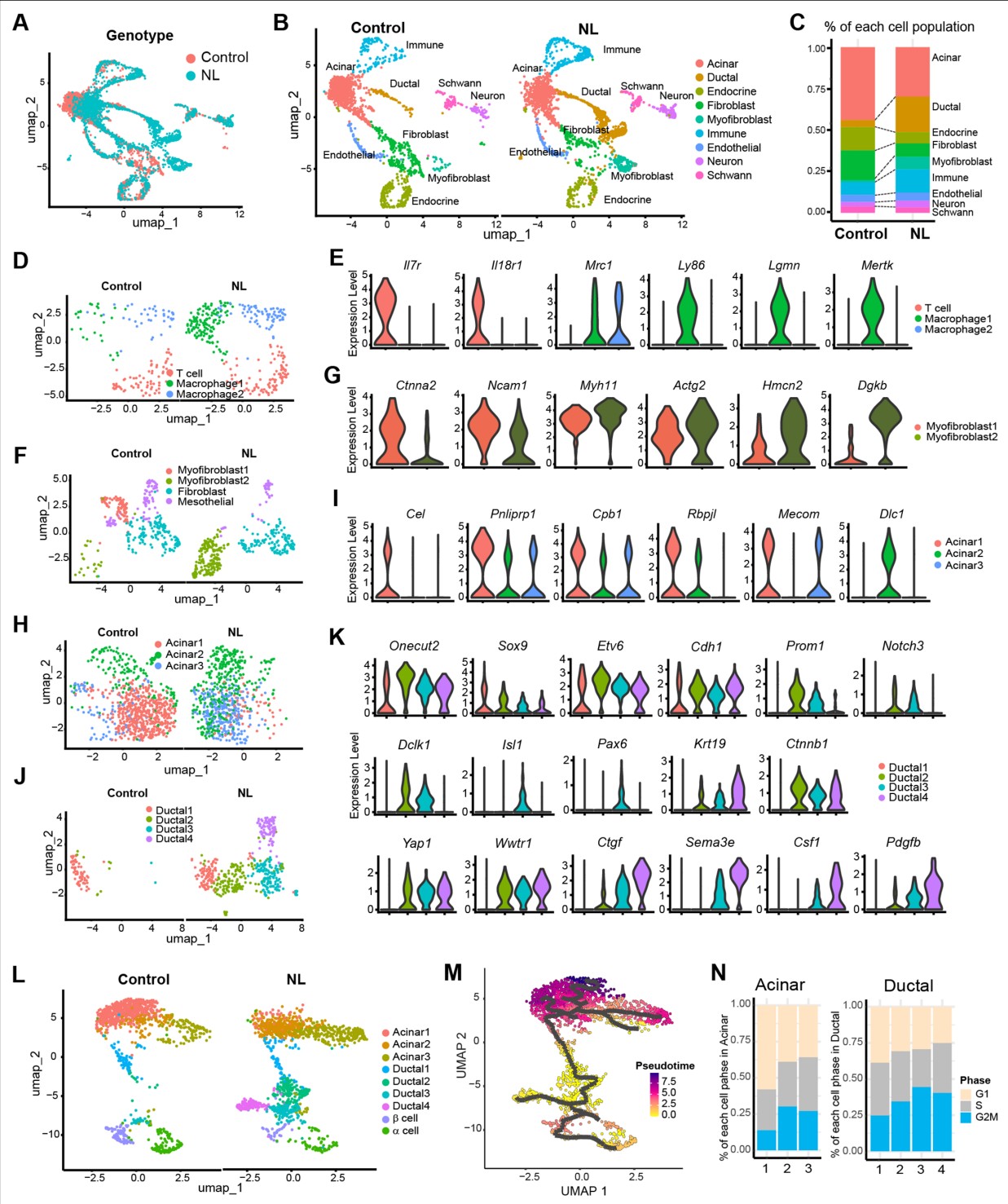

**Figure 6.** Single-nucleus RNA-seq analysis of control and NL mutant mice pancreas. (**A**) Uniform manifold approximation and projection (UMAP) of all single nuclei present in control and NL mice pancreas that passed the QC steps. (**B**) UMAP with cell type annotation for each identified cluster in control and NL pancreas, respectively. (**C**) Percentage composition of each cell type in control and NL pancreas, respectively. (**D**) UMAP of sub-clustered immune cells in control and NL pancreas, respectively. (**E**) Violin plot of expression level of indicated genes in each sub-cluster shown in (**D**), with control and NL pancreas combined. (**F**) UMAP of sub-clustered fibroblasts and myofibroblasts in control and NL pancreas, respectively. (**G**) Violin plot of expression level of indicated genes in each sub-cluster shown in (**F**). (**H**) UMAP of sub-clustered acinar cells in control and NL pancreas, respectively. (**I**) Violin plot of expression level of indicated genes in each sub-cluster shown in (**H**). (**J**) UMAP of sub-clustered ductal cells in control and NL pancreas, respectively. (**K**) Violin plot of expression level of indicated genes in each sub-cluster shown in (**J**). (**L**) UMAP of sub-clustered pancreatic cells (acinar,

*Figure 6 continued on next page*

*Figure 6 continued*

ductal, and endocrine) in control and NL pancreas, respectively. (**M**) UMAP of single-cell RNA trajectory across the cell clusters shown in (**L**), with control and NL pancreas combined. Ductal 2 cell cluster was set as the root. (**N**) Percentage of cells in G1, S, and G2M phase in each acinar and ductal sub-clusters (control and NL pancreas combined).

The online version of this article includes the following figure supplement(s) for figure 6:

**Figure supplement 1.** Single nuclei RNA-seq analysis of control and NL mutant mice pancreas.

architecture and similar endocrine cell mass were observed between control and ML pancreases through H&E staining (*Figure 8B*). To confirm the β-cell-specific *Lats1&2* deletion efficiency, we performed western blot analysis and were unable to detect LATS1&2 protein in islets of ML mice (*Figure 8C*). To further test the influence of *Lats1&2* deficiency on β-cell function, we performed glucose tolerance tests (GTT). Results indicate that ML mice displayed a normal GTT profile when compared to control mice (Ctrl, *n* = 6; ML, *n* = 5) (*Figure 8D*).

Next, we tested whether *Lats1&2* is required for embryonic β-cells. To this end, we injected the pregnant mice carrying Ins1$^{CreER}$*Lats1&2$^{fl/fl}$*Rosa26$^{YFP}$ offspring with Tamoxifen at E12 (*Figure 8E*). The Tamoxifen treatment induced the majority Insulin+ β-cells to express YFP at P1, showing an efficient gene deletion by this procedure (*Figure 8F*). However, the *Lats1&2* deletion did not affect pancreatic architecture and endocrine pancreas morphology (*Figure 8F*). Together, these findings suggest that *Lats1&2* are dispensable for pancreas β-cell proliferation and function.

## Discussion

Recent studies have revealed that the Hippo signaling pathway and its effectors are essential for pancreatic development and function. Multiple Hippo pathway genes have been investigated through specific deletion at embryonic and adult stages using different pancreatic-specific Cre lines in genetically engineered mice models (*Gao et al., 2013*; *George et al., 2012*; *Braitsch et al., 2019*; *Liu et al., 2019*). With the early embryonic deletion of *Lats1&2* using *Pdx1*-early Cre, the developing pancreas loses cell polarity and subsequent undergoes failure of epithelial expansion/branching, indicating that early pancreatic morphogenesis requires a properly functioning Hippo signaling pathway. However, the exact role of the Hippo signaling pathway in endocrine lineage specification has been unclear. Using mice with *Neurog3*-driven Cre, we have revealed that the tight regulation of the Hippo pathway is required for endocrine specification and differentiation. Loss of *Lats1&2* in endocrine progenitor cells blocks their further differentiation, resulting in much smaller pancreatic islets and fewer hormone-producing cells. Further deletion of *Yap1/Wwtr1* rescues the observed endocrine defects in the *Lats1&2* null pancreas, suggesting that YAP1/TAZ transcriptional activities must be tightly controlled by LATS1&2 for endocrine lineage development. Our findings, therefore, expand our understanding of the physiological functions of the Hippo pathway (*Figure 9*).

In line with our histology analysis, we found differences in acinar cells between the control and NL pancreas from our snRNA-seq analysis. Compared to control acinar cells which express high levels of digestive enzyme encoding genes, the most abundant acinar population in the NL pancreas has more cells in a proliferation state. These data suggest that developing acinar cells are responsive to signals from immune and fibroblast cells in their microenvironment. In line with our observation of an increased number of ductal cells in the NL pancreas, snRNA-seq also revealed a significantly larger number of ductal cells, including endocrine progenitors, suggesting that the loss of Hippo signaling control leads to the expansion of both populations. However, it is unclear whether YAP1/TAZ activation is required for the expansion of both populations during pancreas development. High level of YAP1/TAZ in endocrine progenitor cells impedes their further differentiation into mature endocrine cells, resulting in a new population.

Accumulating evidence has shown that Hippo plays dominant roles in the developing exocrine pancreas, but not in the developing endocrine pancreas (*Gao et al., 2013*; *George et al., 2012*; *Duvall et al., 2022*). First, deleting *Mst1&2* with an inducible acinar cell-specific Cre or overexpressing YAP1 in the pancreas reproduced the *Mst1&2* whole pancreas epithelial knockout phenotype – acinar atrophy, ductal expansion, and pancreatitis-like phenotype (*Gao et al., 2013*). In addition, YAP1/TAZ were undetectable in both embryonic endocrine cells and adult endocrine cells (*George et al., 2012*; *Duvall et al., 2022*). Overexpression of *Neurog3* silenced *Yap1/Wwtr1* at the transcriptional level

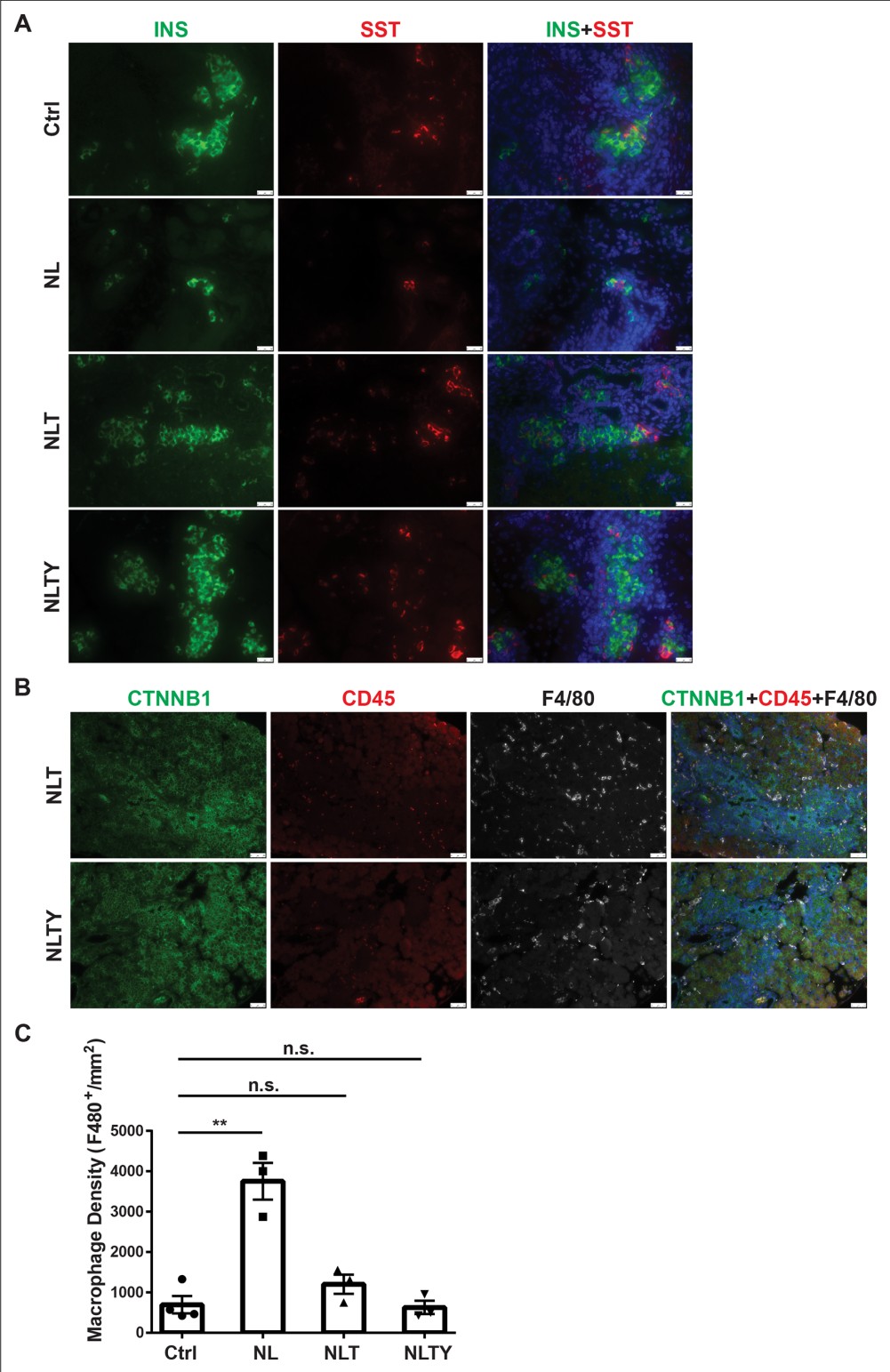

**Figure 7.** Removal of *Yap1/Wwtr1* rescued the defect in endocrine specification and differentiation in NL pancreas. (**A**) Co-staining of Insulin (INS) and Somatostatin (SST) showed that only small cell clusters were positive for INS or SST in NL pancreas, whereas the islets in NLT and NLTY pancreases were INS- or SST-positive, more closely resembling the control islets. Scale bar: 25 μm. (**B**) The numbers of immune cells were much less in both NLT and NLTY pancreases compared to NL pancreas. (**C**) Quantification analysis showed that while the macrophage density of NL pancreas was significantly higher compared to control pancreas, there was no significant difference in

*Figure 7 continued on next page*

*Figure 7 continued*

macrophage density between control and both NLT and NLTY pancreases (Ctrl, *n* = 4 biological replicates; NL, *n* = 3 biological replicates; NLT, *n* = 3 biological replicates; NLTY, *n* = 3 biological replicates). n.s. p > 0.05; **p < 0.01. Scale bar: 50 µm.

The online version of this article includes the following figure supplement(s) for figure 7:

**Figure supplement 1.** Removal of YAP1/TAZ rescued the defect in endocrine differentiation in NL pancreas.

(*Duvall et al., 2022*). Consistent with these findings, we also found that embryonic Neurog3+ cells are negative for YAP1. However, newly formed Neurog3+ cells still exhibit YAP1 expression, but mainly in the cytosol, suggesting that YAP1 is controlled at the post-translational level, most likely by LATS1&2 the canonical Hippo pathway. Our data showed that Neurog3 alone is not sufficient to shut off YAP1 expression. It requires active LATS1&2 to facilitate YAP1/TAZ sequestration outside of cell nuclei. Deletion of *Lats1&2* in Neurog3-expressing cells disrupts ability of Neurog3 to suppress *Yap1/Wwtr1* expression. *Schreiber et al., 2021* have mapped Neurog3 genome occupancy in the human pancreatic endocrine cells and found TEAD expression decreased when NEUROG3 was present. This suggests that YAP1 may regulate its own expression. Further investigation of YAP1 autoregulation and how Neurog3 shuts down YAP1 expression during endocrine differentiation is warranted.

During endocrine pancreas formation, *Neurog3* has to achieve high expression levels for endocrine commitment (*Wang et al., 2010*; *Magenheim et al., 2011*). High *Neurog3* initiates a stepwise differentiation process including EMT and delamination of differentiating endocrine cells. The new inclusive model proposed by *Sharon et al., 2019*, using single-cell sequencing and immunostaining, suggests that islets form in a budding process. Endocrine progenitor cells with a sufficiently high level of Neurog3 expression may leave the epithelial cord, but remain attached to the cord. Sharon et al. suggested that differentiating endocrine precursors do not undergo EMT, but still express E-cadherin (CDH1) throughout the differentiation process. However, they observed downregulation of CDH1 expression during endocrine lineage formation, as previously found (*Gouzi et al., 2011*). 'Leaving the cord' or 'delamination' has been suggested to follow *Neurog3* expression. How Neurog3 mediates CDH1 downregulation and facilitates this process is unclear. We had similar observations in that Neurog3-expressing YFP+ cells were connected to epithelial cords but with lower CDH1 staining in control E16.5 pancreas. However, YFP+ cells in the NL pancreas were connected to the cords and formed buds or sheaths, similar to the control YFP cells, but their CDH1 expression level was as highly as in the epithelial cord. Our findings suggest that high YAP1 activity maintains high CDH1 expression, irrespective of *Neurog3* expression. TEAD1 (*Cebola et al., 2015*) and YAP1 have been found to bind to the *CDH1* promoter region, suggesting that YAP1 may maintain CDH1 expression. However, it is unclear whether loss of YAP1 expression is sufficient to downregulate CDH1 expression in the endocrine lineage. Nevertheless, our findings suggest that an active Hippo pathway is required for the downregulation of CDH1 in *Neurog3*-expressing endocrine progenitor cells.

Notch and Hippo signaling pathways are tightly regulated to govern endocrine lineage determination (*Magenheim et al., 2011*; *Cebola et al., 2015*). High Notch signaling without Hippo generates high HES1 and YAP1, which cooperate to suppress *Neurog3* expression, leading to ductal cell fate. Low Notch signaling with Hippo activity results in *Neurog3* expression mediated by SOX9. The role of SOX9 in endocrine lineage differentiation is complex. SOX9 expresses in the Notch-responsive bipotent progenitors and committed ductal cells. It is suppressed by Neurog3 in committed endocrine progenitors (*Mamidi et al., 2018*). We found that newly born Neurog3+ cells sequestered YAP1 out of nuclei by the Hippo pathway. When Neurog3+ cells grow out of the epithelial cord, YAP1 expression is completely turned off. Loss of *Lats1&2* results in high YAP1 in the nuclei of newborn Neurog3+ cells, which blocks Neurog3's ability to turn on endocrine lineage gene expression. Mamidi et al. have shown that overexpression of YAP1 in progenitor cells can expand the ductal compartment (*Magenheim et al., 2011*). In this setting, *Neurog3* has not been turned on and SOX9 is high, implying that ductal expansion is probably mediated by both SOX9 and YAP1. In our NL mice, *Neurog3* and Cre-recombinase expression are induced simultaneously. Subsequently, *Lats1&2* are deleted and result in YAP1 activation. In these mutant cells, Neurog3 has already turned off SOX9 expression. However, the ductal marker KRT19 is as high as in the ductal cells, suggesting that KRT19 is most likely regulated by YAP1, not by SOX9. Interestingly, a TEAD1- (*Yang et al., 2020*) and YAP1- (*Nipper et al., 2024*) binding peak was found in *KRT19* gene promotor region. Although the mutant cells have high CDH1

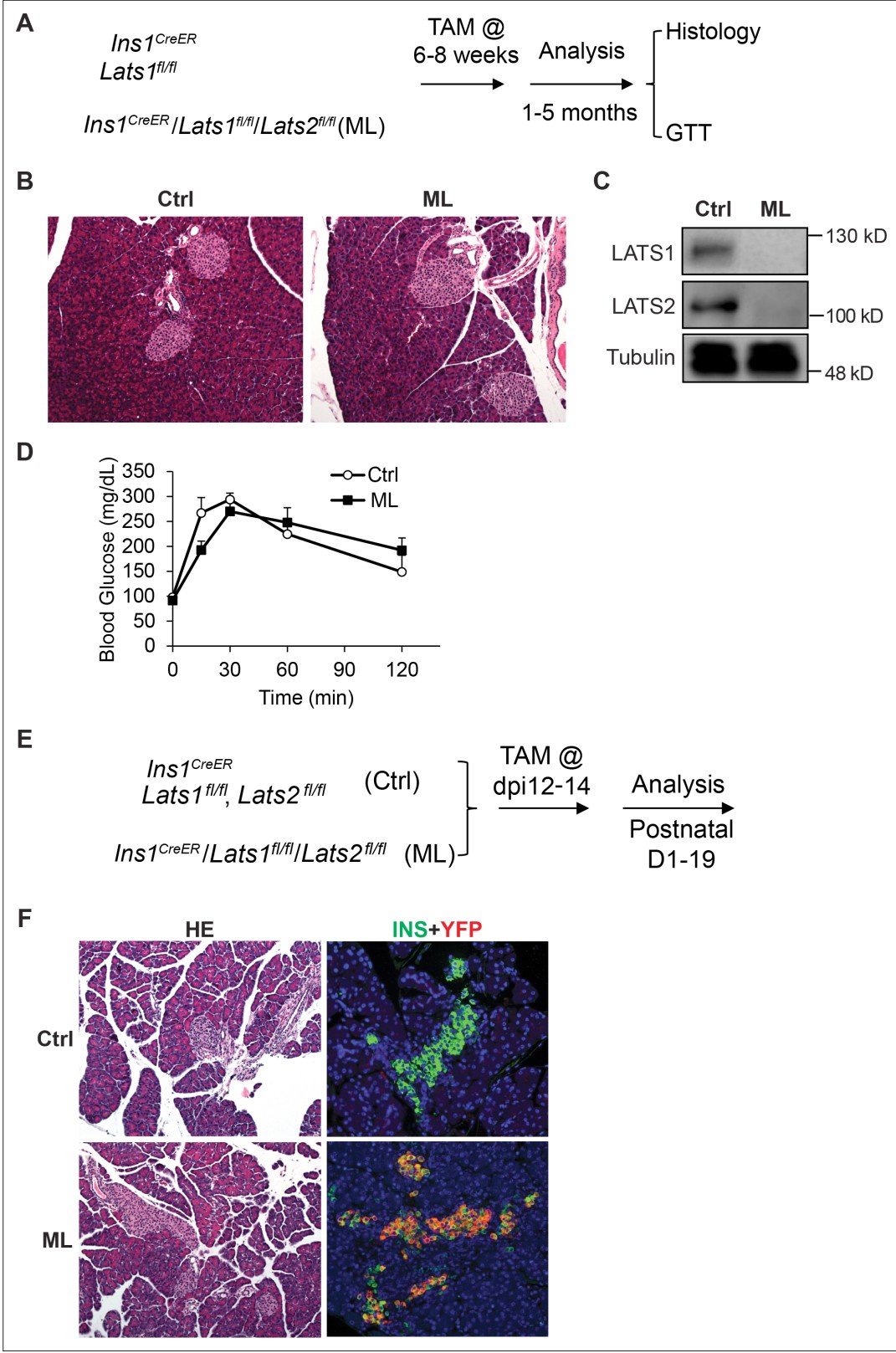

**Figure 8.** *Lats1&2* are dispensable for pancreas β-cell proliferation and function. (**A**) Schematic strategy of deleting *Lats1&2* in adult β-cells using Ins1^CreER to generate ML mice. (**B**) H&E staining showed normal pancreatic architecture and similar pancreatic islet size between control and ML pancreases. (**C**) LATS1&2 protein levels were reduced in pancreatic islets of ML mice. (**D**) The ML mice showed a normal glucose tolerance test (GTT) compared

*Figure 8 continued on next page*

*Figure 8 continued*

with control mice (Ctrl, *n* = 6 biological replicates; ML, *n* = 5 biological replicates). (**E**) Schematic strategy of deleting *Lats1&2* in embryonic β-cells. (**F**) H&E staining showed normal pancreas structure between control and ML mice. Most YFP+ cells expressed INS in P1 ML pancreas.

The online version of this article includes the following source data for figure 8:

**Source data 1.** The original raw unedited blots for LATS1, LATS2, and TUBULIN demonstrated in *Figure 8*, as well as a figure of the uncropped blots with the relevant bands clearly labeled.

and KRT19 expression, and low SOX9 expression, they still leave the epithelial cord and form the sheath, suggesting that 'leaving the cord' is mediated by Neurog3. However, high YAP1 activity in the mutant cells does not support *Neurog3* expression, resulting in no continuous growth of the sheath. Rather, the bulged-out cells return to the epithelial cord. Interestingly, SOX9 expression remains low in these mutant cells, even if they return to simple epithelial cells.

Using human embryonic stem cells, Rosado-Olivieri et al. have showed that overexpression of constitutively active *YAP1* impairs endocrine differentiation while inhibition of *YAP1* can generate improved insulin-secreting cells (*Shih et al., 2012*). This is consistent with our in vivo findings. Interestingly, others have found that overexpression of an active form of YAP1 greatly induces β-cell proliferation in adult human islets (*Duvall et al., 2022*; *Seymour et al., 2007*). Our genetic models indicate that in adult β-cells, *Yap1* gene has been silenced at the transcriptional level. Loss of *Lats1&2* has no effect on β-cell function. Direct increase of YAP1 expression may be required for expansion of β-cells in the adult pancreas. Thus, manipulating YAP1 level to increase β-cell number depends on cell types and developmental stage.

By acinar cell-specific deletion of *Lats1&2*, we showed that PSC activation precedes macrophage activation and infiltration, resulting in extensive fibrosis (*Liu et al., 2019*). In our NL pancreas, some PSC activation is visible, but macrophage infiltration is very distinct even at E16.5 and becomes more conspicuous at P1. Braitsch et al.'s study in which they deleted *Lats1&2* in the very early stage of pancreas development also had similar observations (*Braitsch et al., 2019*), that is the CD45+ cells surrounded the E11.5 mutant pancreas and infiltrated the E14.5 mutant pancreas. Our snRNA-seq analysis showed that the new ductal population in NL pancreas express high level of Csf1, which may lead to recruit macrophages. The tissue-resident macrophages have been suggested to seed the tissue during embryonic development from the blood island of yolk sac and fetal hematopoietic cells in the liver (*Nipper et al., 2024*; *Rosado-Olivieri et al., 2019*). Given the well-known role of Hippo pathway in organ size regulation, it is possible that the abundance of tissue-resident macrophage, including microglia, which are derived from yolk sac myeloid progenitors, is linked to organ

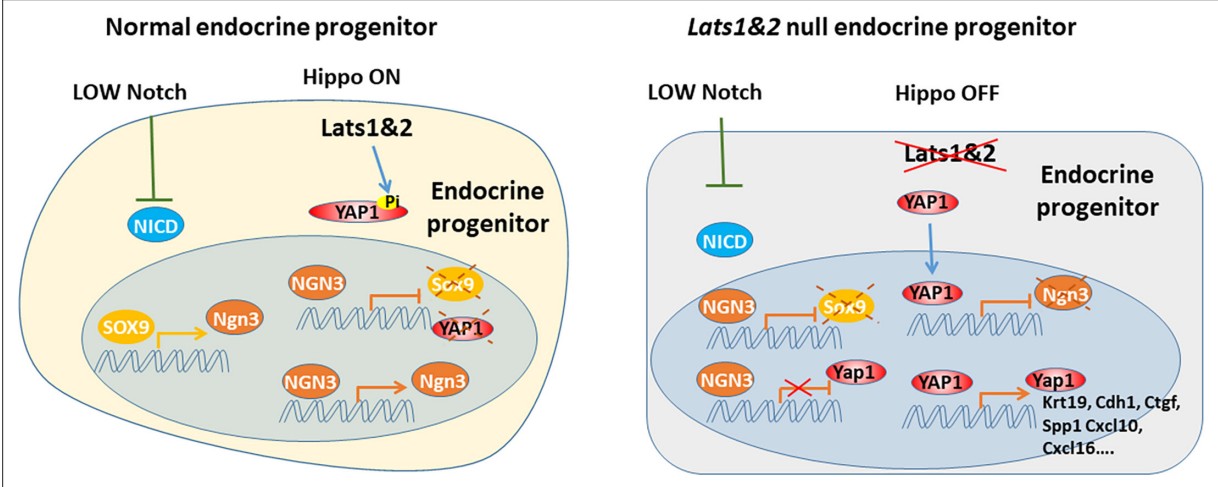

**Figure 9.** During endocrine progenitor specification, the Hippo pathway is required to sequester YAP1 protein in the cytosol and allow Neurog3 protein to positively regulate its own expression and suppress Sox9 and Yap1 expression. Loss of Lats1&2 leads to YAP1 activation which suppresses Neurog3 expression and induces the expression of YAP1 targets.

**Table 1.** Primers for genotyping and qRT-PCR.

| Gene name | Forward | Reverse |
| --- | --- | --- |
| *Lats1* | GCGATGTCTAGCCCATTCTC | GGTTGTCCCACCAACATTTC |
| *Lats2* | AGCCTGACAACATACTCATCG | AATCCAGTGCAGAGGCCAAA |
| *Yap1* | TACTGATGCAGGTACTGCGG | TCAGGGATCTCAAAGGAGGAC |
| *Wwtr1* | GAAGGTGATGAATCAGCCTCTG | GTTCTGAGTCGGGTGGTTCTG |
| *Ankrd1* | TAATCGCTCACAATCTGTTGACA | GCCTCTCACCTTCCGACCT |
| *Ctgf* | GGCCTCTTCTGCGATTTCG | GCAGCTTGACCCTTCTCGG |
| *Cyr61* | TCCTCAGTGAGTTGCCCTC | CCCACCTAAGAGCCTCAGG |
| *Spp1* | AGCAAGAAACTCTTCCAAGCAA | GTGAGATTCGTCAGATTCATCCG |
| *Cxcl10* | CCAAGTGCTGCCGTCATTTTC | GGCTCGCAGGGATGATTTCAA |
| *Cxcl16* | CCTTGTCTCTTGCGTTCTTCC | TCCAAAGTACCCTGCGGTATC |
| *Tgfb1* | CCACCTGCAAGACCATCGAC | CTGGCGAGCCTTAGTTTGGAC |
| *Amy* | TTGCCAAGGAATGTGAGCGAT | CCAAGGTCTTGATGGGTTATGAA |
| *Ptf1a* | TCCCATCCCCTTACTTTGATGA | GTAGCAGTATTCGTGTAGCTGG |
| *Cpa1* | CAGTCTTCGGCAATGAGAACT | GGGAAGGGCACTCGAACATC |
| *Hnf1b* | AGGGAGGTGGTCGATGTCA | TCTGGACTGTCTGGTTGAACT |
| *Sox9* | AGTACCCGCATCTGCACAAC | ACGAAGGGTCTCTTCTCGCT |
| *Krt19* | GGGGGTTCAGTACGCATTGG | GAGGACGAGGTCACGAAGC |
| *Ins1* | CACTTCCTACCCCTGCTGG | ACCACAAAGATGCTGTTTGACA |
| *Ins2* | GCTTCTTCTACACACCCATGTC | AGCACTGATCTACAATGCCAC |
| *ChrA* | ATCCTCTCTATCCTGCGACAC | GGGCTCTGGTTCTCAAACACT |
| *GAPDH* | AGGTCGGTGTGAACGGATTTG | GGGGTCGTTGATGGCAACA |

size and function. Conceivably, the Hippo pathway may influence their migration and proliferation during organ development. This warrants further investigation.

In conclusion, our study suggests that proper Hippo activity is required for the *Neurog3*-driven differentiation program, further expanding our fundamental understanding of Hippo pathway participation in pancreatic endocrine development.

## Materials and methods
### Generation of conditional knockout mice

All animal study protocols were approved by the UT Health San Antonio Animal Care and Use Committee. *Neurog3^Cre* mice (stock number: 005667), *MIP^CreER* (stock number: 024709), and *Rosa26^LSL-YFP* mice (stock number: 006148) were obtained from The Jackson Laboratory. *Neurog3^Cre* mice were kindly provided by Dr. Andrew Leiter and Dr. Seung Kim. *Lats1^fl/fl* and *Lats2^fl/fl* mice were kindly provided by Dr. Randy L. Johnson. *Yap1^fl/fl* and *Wwtr1^fl/fl* mice were kindly provided by Dr. Eric N. Olson. We generated (1) *Neurog3^Cre Rosa26^LSL-YFP Lats1^fl/fl Lats2^fl/+* mice (as Control), (2) *Neurog3^Cre Rosa26^LSL-YFP Lats1^fl/fl Lats2^fl/fl* mice (NL mice), (3) *Neurog3^Cre Rosa26^LSL-YFP Lats1^fl/fl Lats2^fl/+ Yap1^fl/fl Wwtr1^fl/fl* mice (NTY mice), (4) *Neurog3^Cre Rosa26^LSL-YFP Lats1^fl/fl Lats2^fl/fl Yap1^fl/+ Wwtr1^fl/fl* mice (NLT mice), (5) *Neurog3^Cre Rosa26^LSL-YFP Lats1^fl/fl Lats2^fl/fl Yap1^fl/fl Wwtr1^fl/fl* mice (NLTY mice), and (6) *MIP^CreER Rosa26^LSL-YFP Lats1^fl/fl Lats2^fl/fl* mice (ML mice). All offspring were genotyped by PCR of genomic DNA from the toe with primers specific for the *Neurog3^Cre*, *MIP^CreER*, *Rosa26^LSL-YFP*, *Lats1*, *Lats2*, *Yap1*, and *Wwtr1* loci. For the timed mating experiment, male mice were introduced to the cage in the afternoon and removed in the morning on the second day, which was considered gestational day 0.5 (E0.5). The pregnant mice were then used to harvest the embryos at the indicated time. For the tamoxifen-mediated *Lats1&2* deletion in adult

**Table 2.** Primary antibodies list.

| Primary antibody | Company | Catalog # | Dilution | Application |
|---|---|---|---|---|
| alpha-Amylase (AMY) | Sigma-Aldrich | A8273 | 1:500 | IF |
| alpha-SMA (ACTA2) | Santa Cruz Biotechnology | 53142 | 1:250 | IF |
| beta-Catenin (CTNNB1) | BD Biosciences | 610153 | 1:100 | IF |
| CD45 | Biolegend | 103102 | 1:50 | IF |
| KRT19 | DSHB | Troma-III | 1:50 | IF, IHC |
| E-cadherin (CDH1) | Cell Signaling Technology | 3195S | 1:500 | IF |
| F4/80 | Cell Signaling Technology | 70076S | 1:50 | IF |
| GFP (YFP) | Aves | GFP-1020 | 1:500 | IF |
| Insulin (INS) | Abcam | ab7842 | 1:250 | IF |
| ISL1/2 | DSHB | 39.4D5 | 1:50 | IF |
| Ki67 | Biolegend | 652401 | 1:25 | IF |
| Neurogenin 3 (Neurog3) | DSHB | F25A1B3 | 1:50 | IF |
| PDX1 | DSHB | F109-D12 | 1:50 | IF |
| Somatostatin (SST) | Santa Cruz Biotechnology | sc-7819 | 1:100 | IF |
| SOX9 | Cell Signaling Technology | 82630S | 1:50 | IF |
| SPP1 | RD Systems | AF808 | 1:50 | IF |
| TAZ | Sigma-Aldrich | HPA007415-100UL | 1:50 | IF |
| YAP | Cell Signaling Technology | 14074S | 1:100 | IF |
| Tubulin | Proteintech | 11224-1-AP | 1:1000 | WB |
| LATS1 (C66B5) | Cell Signaling Technology | 3477 | 1:1000 | WB |
| LATS2 | Bethyl Laboratories, Inc | A300-479A | 1:1000 | WB |

β-cells, 6- to 8-week-old mice were intraperitoneal (i.p.) injected with 200 mg/kg tamoxifen (TAM, Sigma-Aldrich, T5648) for 5 consecutive days. For the early β-cell *Lats1&2* deletion, female *Lats1&2$^{fl/fl}$* mice and male mice carrying *MIP$^{CreER}$; Lats1&2$^{fl/fl}$* were timed-mated and after 12 days, the pregnant mice were injected with 100 mg/kg tamoxifen once (*George et al., 2015*). PCR was used for validation of knockout alleles (*Table 1*).

## H&E staining, Picrosirius red staining, immunofluorescence, and immunohistochemistry

Pancreases harvested from animals were fixed in 4% paraformaldehyde overnight and then submitted to the Histology Core of the University of Texas Health Science center at San Antonio. The tissues were immersed in serial dilutions of ethanol and xylene and then embedded in the paraffin. The blocks were then cut into 5 µm sections with a microtome. H&E staining was performed at the Histology Core. For Picrosirius red staining, the section slides were first deparaffinized with xylene and washed in serial dilutions of ethanol solutions. Then, the slides were stained with Hematoxylin, 0.1% Fast Green, and Pico-Sirius Red solution (Polysciences, 24901) as previously described (*Trinder et al., 2016*).

For immunofluorescent staining, tissues were deparaffinized, rehydrated, and submerged in 200°C heated R-Universal Epitope Recovery Buffer solution (Electron Microscopy Sciences, Hatfield, PA, 62719-20) for 30 min and then let cool at room temperature for 25 min. Sections were permeabilized using 0.3% tris-buffered saline with Triton (TBST) (0.3% Triton X-100, Acros Organics, Fair Lawn, NJ, 21568-2500) and 0.025% phosphate-buffered saline with Tween20 (PBST) (0.025% Tween20, Fisher Bioreagents, Fair Lawn, NJ, BP337-500) for 4 min each. Sections were subsequently blocked with 10% donkey serum in 0.025% PBST for 35 min at room temperature. Sections were then incubated with primary antibodies diluted in 10% donkey serum in 1× phosphate-buffered

**Table 3.** Secondary antibodies list.

| Secondary antibody | Company | Catalog # | Dilution | Application |
|---|---|---|---|---|
| Alexa Fluor 488-conjugated AffiniPure Donkey Anti-Mouse IgG (H+L) | Jackson ImmunoResearch | 715-545-150 | 1:250 | IF |
| Alexa Fluor 647-conjugated AffiniPure Donkey Anti-Mouse IgG (H+L) | Jackson ImmunoResearch | 715-605-150 | 1:250 | IF |
| Cy3-conjugated AffiniPure Donkey Anti-Mouse IgG (H+L) | Jackson ImmunoResearch | 715-165-150 | 1:250 | IF |
| Alexa Fluor 647-conjugated AffiniPure Donkey Anti-Rabbit IgG (H+L) | Jackson ImmunoResearch | 711-605-152 | 1:250 | IF |
| Cy3-conjugated AffiniPure Donkey Anti-Rabbit IgG (H+L) | Jackson ImmunoResearch | 711-165-152 | 1:250 | IF |
| Alexa Fluor 647-conjugated AffiniPure Donkey Anti-Rat IgG (H+L) | Jackson ImmunoResearch | 712-605-150 | 1:250 | IF |
| Cy3-conjugated AffiniPure Donkey Anti-Rat IgG (H+L) | Jackson ImmunoResearch | 712-165-150 | 1:250 | IF |
| Alexa Fluor 647-conjugated AffiniPure Donkey Anti-Goat IgG (H+L) | Jackson ImmunoResearch | 705-605-147 | 1:250 | IF |
| Cy3-conjugated AffiniPure Donkey Anti-Goat IgG (H+L) | Jackson ImmunoResearch | 705-165-003 | 1:250 | IF |
| Alexa Fluor 647-conjugated AffiniPure Donkey Anti-Guinea Pig IgG (H+L) | Jackson ImmunoResearch | 706-605-148 | 1:250 | IF |
| Alexa Fluor 488-conjugated AffiniPure Donkey Anti-Chicken IgY (IgG) (H+L) | Jackson ImmunoResearch | 703-545-155 | 1:250 | IF |
| Peroxidase-conjugated AffiniPure Goat Anti-Rat IgG (H+L) | Jackson ImmunoResearch | 112-035-003 | 1:250 | IHC |
| Peroxidase-conjugated AffiniPure Goat Anti-Rabbit IgG (H+L) | Jackson ImmunoResearch | 111-035-003 | 1:5000 | WB |

saline (PBS) at 4°C overnight. Then, sections were incubated with fluorescent-tagged Alexa Fluor secondary antibodies (1:250, Jackson ImmunoResearch, West Grove, PA) diluted in 10% donkey serum in 1× PBS for 1 hr at room temperature. Additionally, sections were incubated with diamidino-2-phenylindole dihydrochloride (DAPI) (1:1000, Invitrogen, Carlsbad, CA, P36935) for 4 min at room temperature. Finally, sections were covered with a drop of VectaShield Vibrance Antifade Mounting Medium (Vector Laboratories, Inc, Burlingame, CA, H-1700). All images were captured using Microsystems DMI6000 B microscope and software (Leica Microsystems, Buffulo Grove, IL) and Zeiss LSM510 confocal microscopes. All primary and secondary antibodies used are listed in *Tables 2 and 3*, respectively.

## RNA extraction and reverse transcription real-time PCR analysis

Pancreases from the animal were homogenized in Trizol (Invitrogen, 15596026) using a probe sonicator from Qsonica (25%, 15 s ON, 15 s OFF) immediately after harvest. RNA extraction and reverse transcription real-time PCR (RT-qPCR) were performed as previously described (; *Qin et al., 2018*). Relative expression of mRNA was calculated based on the $^{\Delta\Delta}$Ct method. The primers for quantitative real-time PCR are listed in *Table 1*.

## Western blot analysis

Islets from *Lats1&2$^{fl/fl}$* and *Ins1$^{CreER}$; Lats1&2$^{fl/fl}$* mice were isolated as described (*Qin et al., 2018*). After 3 hr of recovery, 50–60 islets were handpicked into low retention tubes (Eppendorf, Z666548). Total protein was extracted by incubating the islets with Laemmli buffer at 95°C for 5 min and homogenized for another 5 min in a sonication device (Diagenode, B01060001) at 4°C. Protein concentration was determined by the BCA method (Thermo Fischer Scientific, 23225) and 5–10 μg of total protein were loaded onto the 10% sodium dodecyl sulfate–polyacrylamide gel electrophoresis gel for electrophoresis. All antibody information and working concentrations are shown in *Tables 2 and 3*.

## Blood glucose level and GTT

Blood glucose levels of the animal were determined by a glucometer (BIONIME GS550). The GTT was performed on overnight-fasted 5-month post-tamoxifen *Lats1&2$^{fl/fl}$* and *Ins1$^{CreER}$; Lats1&2$^{fl/fl}$* mice. 1.5 g/kg glucose was intraperitoneal injected into animals and blood glucose levels were monitored at indicated time points in the figures.

## Macrophage quantification

All macrophage quantification was performed using the free, open-source program, Fiji (*Schindelin et al., 2012*). First, immunofluorescent staining of DAPI and F4/80 macrophage marker was performed on each pancreas section. Four randomly chosen, representative areas of each mouse pancreas were captured at ×20 objective magnification. Each fluorescent channel of the chosen areas was saved as separate images. Each of these separate channel images were then overlayed using the Image Calculator Tool in Fiji in order to visualize DAPI and F4/80 signal, together. Once combined, using the Channels Tool, each channel can be overlayed or removed as desired. The number of pixels per micrometer was calculated and defined using the Straight Line Tool and Measure Tool in Fiji. This allows for the total area of any given selection to be measured in micrometers squared. Using the DAPI image channel and Threshold tool, a threshold limit was set to 15 for the minimum and 255 for the maximum, and total DAPI-positive area (in micrometers squared) within this threshold parameter was calculated for each embryonic pancreas region. Then, each F4/80+ DAPI+ cell was hand counted within each respective region using the Multi-point tool in Fiji. Finally, the total number of counted F4/80+ macrophage cells was divided by the measured DAPI+ area. This yields the calculated macrophage density of total number of F4/80+ macrophage per micrometer squared in regions of the pancreas. For quantification of total macrophage density per image, the calculated density of each acinar- and ductal-cell region per image were averaged together. All images were captured using Microsystems DMI6000 B microscope and software (Leica Microsystems, Buffalo Grove, IL).

## Statistical analysis

All quantification results in this study except bioinformatic analysis described below were presented as the mean ± standard error of the mean. Statistical analysis was performed by a two-tailed Student's *t*-test. All p-values <0.05 were considered statistically significant. Sample size was not calculated before the experiment. The sample size was determined based on the availability of generated conditional knockout mice to provide at least three biological replicates for a sufficient statistical power.

## Single-nucleus RNA-seq sample preparation

Mouse E17.5 pancreas was flash frozen in liquid nitrogen and minced to smaller chunks on ice with sterile micro-dissection scissors. The finely minced tissues were transferred to a pre-chilled 2 ml Dounce homogenizer containing 2 ml cold 1× Homogenization Buffer (HB) which contained 0.26 M sucrose, 0.03 M KCl, 0.01 M $Mg_2Cl$, 0.02 M Tricine–KOH pH 7.8, and Ultra-Pure Water (Thermo Fisher), with the following added on the day of tissue processing – 1 mM Dithiothreitol (DTT), 0.5 mM Spermidine (Sigma-Aldrich), 0.15 mM Spermine (Sigma-Aldrich), 0.3% NP40, 1× Roche Complete Protease Inhibitor Cocktail, 1 U/µl RNase Inhibitor (Fisher Scientific). The tissue was homogenized on ice by moving a 'loose' pestle up and down 10–15 times to disrupt tissue structure, followed with a 'tight' pestle to achieve a final homogenate. This homogenate was then strained through a 70-µm Flowmi strainer into a 2-ml DNA/RNA LoBind tube and centrifuged at 4°C at 350 rcf for 5 min. After the centrifugation, the supernatant was discarded, and the pellet was washed with 1 ml cold PBS, passed through a 40-µm Flowmi strainer, and centrifuged again. For density gradient separation, the isolated nuclei were resuspended in 400 µl of 1× HB and combined with an equal volume of 50% chilled Iodixanol solution. Subsequent layers of 30% and 40% Iodixanol solutions, diluted in 1× HB, were gently layered under the initial mixture to create a gradient. In the final step, the tube with the nuclei was centrifuged using a pre-cooled bucket-rotor centrifuge at 3000 rcf for 20 min at 4°C. After centrifugation, the top layer was carefully removed, without disrupting the 'fuzzy' white ring of nuclei. A narrow-bore pipette tip was used to collect no more than 200 µl of the nuclei, which were then mixed with an equal volume of 1× HB. Nuclei concentration was determined using a Countess II FL automated cell counter (Thermo Fisher Scientific), adjusting to a final concentration optimal for 10x Genomics Chromium library preparation.

## Single-nucleus RNA-seq data analysis

Base calling was performed using RTA 3.4.4, demultiplexing was performed using cellranger-arc v2.0.1 (Bcl2fastq 2.20.0), and alignment was performed using cellranger-arc v2.0.1 (BWA 0.7.17-r1188). Sequenced reads were aligned to a 10x Genomics provided mouse reference sequence (refdata-cellranger-arc-mm10-2020-A-2.0.0). The subsequent analysis was performed using Seurat

v.5.0.1 in R software v.4.2.0. A quality control step was performed in merged control and mutant samples to remove low-quality cells, including possible cell debris (nFeature <200), cell doublets (nFeature >10,000), and cells with excessive cellular stress (>5% mitochondrial gene expression). The merged/filtered Seurat object was then subjected to a series of standard Seurat commands including 'NormalizeData', 'FindVariableFeatures', 'ScaleData', 'RunPCA', 'FindNeighbors', 'FindClusters', and 'RunUMAP', with 20 PCA dimensions (determined by elbow plot with ndims = 50) and a resolution of 0.2, in order to identify different cell populations present in the dataset. 'FindAllMarkers' function was performed to identify marker gene expression in each cell population identified in the previous step (only.pos = TRUE, min.pct = 0.25, logfc.threshold = 0.25). Each cell population was then assigned to a specific cell type by manually reviewing their corresponding marker gene expression with prior knowledge. Individual gene expression was visualized using 'VlnPlot', 'FeaturePlot', and 'DotPlot' functions in Seurat with default setting. The cell cycle status of the cells was determined using 'CellCycleScoring' command in Seurat with default setting. To analyze each individual cell type, the merged Seurat object was subset to smaller objects only containing cells with specific cell identities. The subset Seurat objects were then subject to similar analysis for re-clustering (including 'FindVariableFeatures', 'ScaleData', 'RunPCA', 'FindNeighbors', 'FindClusters', and 'RunUMAP') followed by further analysis. GSEA as well as individual plotting was performed using ClusterProfiler v4.4.4 in R software. In order to compensate the significant difference in reading depth between Myofibroblast 1 and 2 clusters, fold change of gene expression was calculated using the averaged gene expression of all the cells in each cluster, instead of the raw read counts. In addition, only genes with >25% expression in each cluster were included in the GSEA analysis. A significant enrichment was considered with multiple-test adjusted p-value <0.05.

## Differential expression analysis

To identify differentially expressed genes (DEGs) among the four Ductal sub-clusters, 'FindMarkers' Seurat function was performed to compare each pair of the four sub-clusters. The resulting genes were filtered to identify DEGs defined as log2 fold change >1.5 or <−1.5, and adjusted p < 0.01. A pseudobulk sequencing dataset for each Ductal sub-cluster was created by manually aggregating the gene read counts of all cells in each cluster. The pseudobulk dataset was then passed to DESeq2 v.1.36.0 and ComplexHeatmap v.2.13.2, to perform *K*-means clustering in order to identify expression patterns of the DEGs across the four Ductal sub-clusters. After identification of *K*-means clustering, the information was passed back into Seurat to plot gene expression heatmap at single-cell level. Overrepresentation analysis for lists of genes was performed using the clusterProfiler v.4.4.4 package in R with default settings. A significant enrichment was considered with multiple-test adjusted p-value <0.05.

## Single-cell trajectory analysis

Single-cell trajectory analysis was performed using Monocle3 v.1.3.4 in R software with default setting. The subset Seurat object containing all acinar, ductal, and endocrine cells were passed into Monocle3 to infer their cell-type transition states, by setting Ductal 2 population as the root.

## Acknowledgements

The authors acknowledge Dr. Andrew Leiter and Dr. Seung Kim for kindly providing the *Neurog3*Cre mouse line, Dr. Randy Johnson for kindly providing the *Lats1*fl/fl and *Lats2*fl/fl mouse line, and Dr. Eric N Olson for kindly providing the *Yap1*fl/fl and *Wwtr1*fl/fl mouse line. The authors thank Dr. Guoqiang Gu (Vanderbilt University) and Dr. Yi Xu for their critical comments. High-magnification confocal images were generated in the Core Optical Imaging Facility at UT Health San Antonio. We thank Dr. Michael Kelly and staff at the Single Cell Analysis Facility (Leidos Inc, Frederick, MD) for their assistance with single-cell sequencing. Computational resources of the NIH High Performance Cluster (Biowulf) supported the analysis in this work (http://hpc.nih.gov). Pei Wang is a CPRIT scholar. This work is supported by the Cancer Prevention and Research Institute of Texas (P Wang, R1219), NIDDK (P Wang, R01DK110361), and the funds from the Intramural Research Program of the Center for Cancer Research, National Cancer Institute of the United States (ZIA BC011798 to HEA). The PW group was supported by Cancer Prevention and Research Institute of Texas, the William and Ella Owens Medical Research Foundation, and National Cancer Institute (R21 CA218968, R01 CA237159). Michael Nipper

and Xue Yin were supported by a pre-doctoral fellowship through CPRIT Research Training Award RP 170345; Jun Liu was supported by a post-doctoral fellowship through CPRIT Research Training Award RP140105. The Core Optical Imaging Facility is supported by UT Health San Antonio and NIH-NCI P30 CA54174. The funders had no role in study design, data collection and analysis, decision to publish, or preparation of the manuscript.

## Additional information

### Funding

| Funder | Grant reference number | Author |
|---|---|---|
| Cancer Prevention and Research Institute of Texas | R1219 | Pei Wang |
| National Institute of Diabetes and Digestive and Kidney Diseases | R01DK110361 | Pei Wang |
| Cancer Prevention and Research Institute of Texas | RP170345 | Michael H Nipper Xue Yin |
| National Cancer Institute of the United States | ZIA BC011798 | H Efsun Arda |
| William and Ella Owens Medical Research Foundation, and National Cancer Institute | R21 CA218968 | Pei Wang |
| William and Ella Owens Medical Research Foundation, and National Cancer Institute | R01 CA237159 | Pei Wang |
| CPRIT Research Training | RP140105 | Jun Liu |

The funders had no role in study design, data collection, and interpretation, or the decision to submit the work for publication.

### Author contributions

Yifan Wu, Kunhua Qin, Data curation, Formal analysis, Methodology, Writing – original draft, Writing – review and editing; Yi Xu, Formal analysis, Visualization, Methodology, Writing – original draft; Shreya Rajhans, Data curation, Formal analysis, Visualization, Methodology, Writing – original draft; Truong Vo, Methodology; Kevin M Lopez, Data curation, Formal analysis, Writing – review and editing; Jun Liu, Data curation, Formal analysis, Methodology, Writing – review and editing; Michael H Nipper, Data curation, Writing – review and editing; Janice Deng, Xue Yin, Logan R Ramjit, Data curation; Zhenqing Ye, Yu Luan, Formal analysis; H Efsun Arda, Formal analysis, Supervision, Funding acquisition, Writing – original draft, Project administration, Writing – review and editing; Pei Wang, Conceptualization, Data curation, Formal analysis, Supervision, Funding acquisition, Investigation, Visualization, Methodology, Writing – original draft, Project administration, Writing – review and editing

### Author ORCIDs

Yifan Wu ⓘ https://orcid.org/0000-0002-8124-8651
Kevin M Lopez ⓘ https://orcid.org/0000-0002-3163-4634
H Efsun Arda ⓘ https://orcid.org/0000-0002-5294-2521
Pei Wang ⓘ https://orcid.org/0000-0003-2373-7315

### Ethics

All animal procedures have been approved by the Institutional Animal Care and Use Committee (IACUC) at The University of Texas Health San Antonio (protocol #20130073).

### Decision letter and Author response

Decision letter https://doi.org/10.7554/eLife.84532.sa1

Author response https://doi.org/10.7554/eLife.84532.sa2

## Additional files

### Supplementary files
• MDAR checklist

### Data availability

snRNA-Seq data presented in this publication have been deposited in NCBI's Gene Expression Omnibus (*Edgar et al., 2002*) and are accessible through GEO Series accession number GSE262400. All of the software used in this study, including Rstudio v.4.2.0, Seurat v.5.0.1, ClusterProfiler v4.4.4, DESeq2 v.1.36.0, and ComplexHeatmap v.2.13.2, are public available for download, except cellranger-arc v2.0.1 is commercially provided by 10X genomics.

The following dataset was generated:

| Author(s) | Year | Dataset title | Dataset URL | Database and Identifier |
|---|---|---|---|---|
| Yifan W, Kunhua Q, Yi X, Shreya R, Kevin L, Jun L, Michael N, Janice D, Xue Y, Logan R, Zhengqing Y, Yu L, Arda HE, Wang P | 2024 | Hippo pathway-mediated YAP1/TAZ inhibition is essential for proper pancreatic endocrine specification and differentiation | http://www.ncbi.nlm.nih.gov/geo/query/acc.cgi?acc=GSE262400 | NCBI Gene Expression Omnibus, GSE262400 |

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
