## [Editor Report]

This study presents the essential role that hippo pathway plays in proper pancreatic endocrine specification and differentiation. The authors demonstrate that the deletion of the Lats1 and 2 kinases (Lats1&2) in endocrine progenitor cells of developing mouse pancreas with Ngn3-Cre blocked endocrine progenitor cell differentiation and specification, resulting in reduced islets size and disorganized pancreas at birth, but no effects were observed when deleting them in β cells. These results show for the first time that Hippo pathway-mediated YAP1/TAZ inhibition in endocrine progenitors is a prerequisite for endocrine specification and differentiation.

---

## [Decision Letter]

[Editors' note: this paper was reviewed by Review Commons.]

Thank you for submitting your article "Hippo pathway-mediated YAP1/TAZ inhibition is essential for proper pancreatic endocrine specification and differentiation" for consideration by *eLife*. Your article has been reviewed by 3 peer reviewers at Review Commons, and the evaluation at *eLife* has been overseen by a Reviewing Editor and Didier Stainier as the Senior Editor.

Based on the previous reviews and the revisions, the manuscript has been improved but there are some issues that need to be addressed, as outlined below:

The phenotype is clear although the authors don't dig deeper into the mechanism:

1. Is indeed YAP directly regulating Ngn3 expression or is Ngn3 directly regulating hippo? To my knowledge no one has done this. Maybe ChIP seq or at least ChIP-PCR will answer this.

2. The phenotype is due to lack of delamination of endocrine cells from the trunk? inhibition of EMT?

Although the authors look at CDH1 expression, they don't look anything other than IF and include statements (without experiments) such as:

"These data suggest that an early effect of Lats1&2 deletion in NGN3+ cells is to activate KRT19 expression, but not *SOX9* expression, further indicating that KRT19 expression is not controlled by *SOX9*, but instead by YAP1"

(This is just based on *Sox9* and Krt19 IF…)

3. Is Lats KO in Ngn3 cells forcing the bipotent trunk progenitor to exclusively differentiate into ducts?

– Why is acinar differentiation affected if they floxed Lats by using Ngn3Cre? Is a paracrine effect? Cell to cell communication?

– No proof of Lats inactivation is showed in Ngn3Cre mice

– The authors talk about PSC activation and paracrine communication but don't check anything to indeed validate his point.

– In general, the analysis can be extended. Unfortunately, the authors base all the statements by looking at stainings (not even RNAseq to look at DE genes).

I would recommend a deeper analysis of the animal models to support the statements. Although we believe the findings are interesting, there is need for some major analysis prior to publication.

---

## [Author Response]

General Statements

The goal of our study is to understand the role of the Hippo pathway during endocrine pancreas development. Using our genetically engineered mouse models, we reveal the pivotal role of Hippo signaling in differentiation and specification of endocrine progenitor cells. Genetic deletion of *Lats1/2* in pancreatic endocrine progenitor cells (NL model: *Ngn3^Cre^Lats1^fl/fl^Lats2^fl/fl^*) shows significant endocrine development defects. Using our rescue mouse model containing *Lats1/2* and *Yap1/Taz* quadruple-null endocrine progenitors (NLTY: *Ngn3^Cre^Lats1^fl/fl^Lats2^fl/fl^Yap1^fl/fl^Taz^fl/fl^*), we showed almost complete rescue from developmental defects observed in our NL model. Genetic ablation of *Yap1/Taz* in our NLTY model mimics canonical Hippo kinase cascade-mediated control of YAP1/TAZ by LATS1/2. Uncontrolled YAP1/TAZ due to genetic ablation of *Lats1/2* in our NL model blocked endocrine specification and differentiation, suggesting the importance of intact Hippo signaling for endocrine development. We value all the comments from reviewers and revised our manuscript to strengthen our findings by: reperforming several immunostaining experiments to obtain higher quality images, clarified terminology and explanations throughout our manuscript, and updated the details on the methods we used to obtain our results. Further details on our revisions are explained below.

Point-by-point description of the revisionsReviewer #1CROSS-CONSULTATION COMMENTS– All reviewers have similar doubts on quality and quantification of the IHC analyses, so I would agree to give a chance to the authors for a revision, and it is probably doable in between 3 and 6 months (I wrote >6 before).

In our fist submission, our full PDF file did not use the high-resolution figures due to large file size of some images. For re-submission, we uploaded our original, high-resolution figures. We also re-performed several immunostaining experiments and took confocal images to increase the quality of our figures. Please see below for detailed response.

Reviewer #1 (Evidence, reproducibility, and clarity (Required)):Summary:In this study, Wu et al. confirmed the prominent cross talk of the Hippo pathway in β-cell development, namely of Ngn3-YAP and β-cell maturation. Deletion of the Hippo kinases Lats1 and 2 in Ngn3-expressing endocrine progenitor cells activated YAP1/TAZ transcriptional activity and reduced islets size and morphology and induced pancreas inflammation. This is in line with the previously established role of YAP/TAZ as regulator of viral response pathwaysIn contrast, deletion of Lats1&2 later in development had no effect on β-cell function and morphology. The authors conclude that Hippo pathway-mediated YAP1/TAZ inhibition in endocrine progenitors is a prerequisite for β-cell maturation.– While the effect of Lats1 and 2 in β-cell development has never been investigated, the outcome of the study is largely confirmative; namely YAP's necessity to switch off whenever an endocrine cell is formed while it also balances inflammation.The study depends on cell-specific Lats1 and 2-KO mouse models and in-vitro assessments of mechanisms how loss of LATS leads to β-cell derangement at the time of ngn3 expression and how the inflammatory pathway is activated are missing. Observations are mostly based on stainings of quite low quality and it is unclear how authors performed quantitative evaluations and how many cells from how many mice were counted. Figures show high background, e.g. CD45 in Figure 3 which would make a robust evaluation impossible. Also YAP staining, usually well-expressed in ductal cells, shows low quality and high background. Usually the antibody is well-know for its weak performance in fluorescence and should only be used with chromogenic labels.

To show the co-localization of multiple proteins, immunofluorescence staining is necessary.

We added and described, in detail, how we performed macrophage quantification in the Methods section.

Reviewer #2 (Evidence, reproducibility, and clarity (Required)):Summary:Wu et al. examined the roles of Hippo pathway mediated YAP1/TAZ inhibition in the development stages of endocrine specification and differentiation in vitro and in vivo. This study concluded Hippo pathway-mediated YAP1/TAZ inhibition in endocrine progenitors is a prerequisite for endocrine specification and differentiation. The present study is conducted by solid experiment in some parts, however, there are several major concerns as follows.Major comments:– Authors concluded that proper Hippo activity was required for the Ngn3 driven differentiation program, further expanding our fundamental understanding of Hippo pathway participation in pancreatic endocrine development. This sentence is too vague.Previous study has already reported that Ngn3 expression and Yap loss occur in parallel within the same cell during development of the endocrine pancreas (Mol Endocrinol Baltim Md. 2015;29: 1594-607. doi:10.1210/me.2014-1375).

As reviewer pointed out, the previous study only reported that *Ngn3* expression and YAP1 loss occur in parallel within the same cells, and NGN3 can turn off *Yap1* transcription in cell culture setting. However, our study showed that NGN3 alone is not sufficient to turn off *Yap1* gene in vivo. Using our genetic model, we revealed that the Hippo pathway controling the nuclear localization of YAP1 is essential at the initiation of endocrine differentiation and allows YAP1 to be turned off at the transcriptional level.

Our rescue model (NLTY mice) confirmed that the endocrine development defects observed in NL pancreas are caused by YAP1/TAZ, suggesting the necessity of controlling YAP1/TAZ by the Hippo pathway during endocrine development. Failure to sequester YAP1/TAZ outside of nuclei by the Hippo pathway in *Ngn3*-expressing endocrine progenitors halts the development of these cells.

– In the present study, removal of YAP1/TAZ rescued the defect in endocrine specification and differentiation in LATS1/2-null pancreas. These results indicate that both of LATS1/2 and YAP/TAZ are not essential for the normal endocrine specification and differentiation. Furthermore, these results suggest that Hippo pathway-mediated YAP1/TAZ inhibition is also unnecessary for proper pancreatic endocrine specification and differentiation. Additionally, these results suggest that just a deletion of YAP/TAZ is sufficient for endocrine specification and differentiation. How is the endocrine specification and differentiation in Ngn3creyap1fl/flTazfl/fl mice?

Our NL model (deletion of *Lats1/2* in *Ngn3*-expressing cells) showed endocrine development defects, suggesting that *Lats1/2* are essential for normal endocrine specification and differentiation. Removal of YAP1/TAZ in *Lats1/2* null cells rescued the endocrine defects of NL pancreas. Genetic removal of YAP1/TAZ mimics Hippo pathway-mediated sequestration/degradation of YAP1/TAZ by LATS1/2. Uncontrolled YAP1/TAZ due to genetic removal of *Lats1/2* blocked the endocrine specification and differentiation. This further demonstrates the necessity for functional *Lats1/2* for inhibition of YAP1/TAZ in *Ngn3*-expressing cells during endocrine development. *Ngn3^Cre^Yap1^fl/fl^Taz^fl/fl^* mice have no endocrine defects, which is consistent with previous reports that *Ngn3* expression and YAP1 loss occur in parallel within the same cells. Thus, we did not present this result.

­ In Figure 2B and 3B, YFP expression (an indicator for LATS1/2 deletion) is more detectable in control compared to LATS1/2 null mice, suggesting that the LATS1/2 expression is more decreased in control mice. Is this true?

YFP expression shows Cre activity which can be used as an indicator for *Lats1/2* deletion. However, the expression level of YFP does not correlate with the expression level of *Lats1/2* because YFP is controlled by the *Rosa26* gene promoter. We did observe a higher level of YFP expression in endocrine cells compared with non-endocrine cells in NL pancreas. It is possible that the *Rosa26* promoter is more active in endocrine cells. However, this is beyond the scope of our research.

– Please show the expression of YFP, NGN3, and YAP/TAZ in figure 6A to confirm their expression status.

In Figure 6, we intend to show that loss of YAP1/TAZ can rescue NL mice defects at P1 where *Ngn3* expression is low. The NLTY pancreas showed much more endocrine cells compared to NL pancreas, suggesting that *Lats1&2* functions to control YAP1/TAZ. Genetic ablation of YAP1/TAZ in *Ngn3*-expressing endocrine cells equals sequestering of YAP1/TAZ outside of nuclei by LATS1/2. Thus, we did not show expression of YFP, NGN3, and YAP1/TAZ in Figure 6A. Instead, we reperformed immunostaining for Figure 5B and took high-magnification confocal images to show expression pattern of NGN3 and YAP1 in endocrine progenitors.

– Please present the sequential changes of LATS1/2, YAP/TAZ, NGN3 expressions in the control mice and LATS1/2 null mice at least E12.5, E16.5, and P1, to make ii easy to understand them for general readers.

At E12.5, the Hippo pathway plays important roles in pancreas development. See reference 11. *Ngn3* peaks at E15.5 and only expresses in endocrine progenitors. We have published a review paper (ref. 7) “Wu Y, Aegerter P, Nipper M, Ramjit L, Liu J, Wang P. Hippo Signaling Pathway in Pancreas Development. Front Cell Dev Biol. 2021;9: 663906. doi:10.3389/fcell.2021.663906” as a detailed background for this manuscript, including pancreas development, up-to-date publication on Hippo pathway in pancreas development, and a model for YAP1 function in endocrine development. Thus, we did not include this background information in this manuscript.

Minor comments:– Please describe the affiliation of authors (number 3 and 4).

It has been included in the manuscript: ^3^Department of Molecular Medicine ^4^Department of Population Health Sciences.

– In table 1, the details of secondary antibodies were not described.

We have added tables to show all primary and secondary antibodies used in the paper.

Reviewer #2 (Significance (Required)):– Previous studies have already reported that Ngn3 expression and Yap loss occur in parallel manner during development of the endocrine pancreas. The primary aim of the present study is whether the YAP loss is mediated by LATS1/2 in Ngn3 positive cells.

The primary aim of our study is to understand if the Hippo pathway plays an important function in endocrine pancreas development.

– Furthermore, although authors concluded that Hippo pathway (LATS1/2)-mediated YAP1/TAZ inhibition is essential for proper pancreatic endocrine specification and differentiation, the null condition both of LATS1/2 and YAP/TAZ in Ngn3 positive cells provided the normal endocrine specification and differentiation (Figure 6). These findings do not support the conclusion.

The defects in our NL (*Lats1/2* null) pancreas in endocrine development strongly suggest the essential role of Hippo pathway in endocrine development. The null conditions of both *Lats1/2* and *Yap1/Taz* (NLTY mice) mimic the effects of LATS1/2-mediated control of YAP1/TAZ in endocrine progenitors via genetic ablation rather than through the canonical Hippo kinase cascade. This further demonstrates the importance of proper Hippo signaling during endocrine development to inhibit YAP1/TAZ via the Hippo kinase cascade in *Ngn3*-expressing endocrine progenitors.

Reviewer #3 (Evidence, reproducibility, and clarity (Required)):Summary:In this study, Wu et al. use murine Cre-lox model systems to demonstrate that the core Hippo pathway components, Lats1/2, promote pancreatic endocrine specification and differentiation through Yap1/Taz. The roles of the Hippo pathway in mammals are complicated and context-dependent. Prior studies have implicated the core kinase cascade of the Hippo pathway, containing MST_1/2_, LATS1/2 and YAP1/TAZ, in pancreatic cell lineage differentiation and morphogenesis. However, the necessity of the Hippo pathway in the development of the endocrine pancreas in vivo remains unsettled. This study by Wu et al. demonstrates that the Hippo pathway is essential for endocrine progenitor specification and differentiation but not for pancreatic β cell function in mice. Their results are in line with prior studies, but some major issues ensue largely due to the lack of data quantification to substantiate the authors' claims, the use of ambiguous/imprecise terminologies, and making unsubstantiated claims.Major comments:– Immunofluorescence was used to evaluate the expression, co-localization, and subcellular localization of proteins of interest (Figures 2~7). However, except for the estimation of macrophage densities (4D and 6C), none of the other immunofluorescence experiments are accompanied by quantifications and statistical analyses to substantiate the authors' claims. In addition, the red channel in several figure panels (eg. 2A and 4A) was suboptimal making interpretations very difficult. Quantification of immunofluorescence data can be done by manual counting or automated counting using software such as ImageJ or QuPath, followed by statistical analyses to provide objective evidence.

We did not upload high resolution images for Bioxriv publication. We now uploaded the high resolution images. We also repeated a few immunostaining experiments and have taken confocal images to increase clarity.

– The endocrine component constitutes only a portion of the pancreas. The authors use whole pancreases to compare the expression levels of cell type-specific or Yap1 target genes between the knockout and control mice through qPCR. While there may be technical feasibility reasons limiting the direct assessments of gene expression in the endocrine progenitor cells, the caveats of the experiments (eg. inferring cell type-specific gene expression changes from whole organ) should be highlighted along with any inconsistent results. For example, among the three ductal genes tested in Figure 1C, only Krt19 has significantly higher mRNA expression in NL vs control, but the possible reasons for such discrepancy between ductal markers were not discussed. The authors used immunofluorescence, which is only semi-quantitative, to determine that Yap1 protein abundance is increased in Lats1&2 knockout cells. qPCR should also be performed for Yap1 in addition to Yap1 target genes to augment their claim of Yap1 expression being increased. More details on how the relative mRNA expression was computed is also necessary for readers to accurately interpret the results.

As reviewer pointed out that there are technical feasibility reasons limiting the direct assessments of gene expression in the endocrine progenitor cells. We can only quantify gene expression level through RT-qPCR. We have discussed the high level of *Krt19* and unchanged level of *Sox9* in Figure 3B and 3C when we performed immunostaining. The results from the two experiments are consistent. We observed KRT19 positive staining but no *SOX9* staining in *Lats1/2* null *Ngn3*-expressing cells. This result is consistent with RT-qPCR result. We have postulated in the Discussion section that *Krt19* may be directly controlled by YAP1.

Furthermore, *Lats1/2* controls the protein level and localization of YAP1/TAZ, thus immunostaining can show the cellular localization.

In addition, we added the n’s used to perform RT-qPCR experiments to our manuscript and figures. We also added details to provide clarity on calculations made for relative mRNA expression from RT-qPCR experiments in the Methods section.

– The increased immunofluorescent detection of Yap1 in the NL pancreases at E16.5 is interesting but warrants further investigation. Is the increase restricted to endocrine progenitor cells or all endocrine compartments? George et al. (Mol Endocrinol. 2015) used RNA in situ hybridization and found that Yap1 mRNA expression is undetectable in the endocrine pancreas at E16.5, but here the authors observe increased Yap1 protein detected by immunofluorescence in the pancreases of animals with Lats1&2 knockout at E16.5. Although the authors speculate on a possible mechanism that may explain the discrepancies, it is important to evaluate whether the knockout really results in reactivation of Yap1 transcription and whether Yap1 auto-regulates on the transcriptional level.

The increase of YAP1 staining is restricted to endocrine progenitor cells and blocks the endocrine differentiation. The endocrine cells in NL pancreas are escapers of *Lats1/2* deletion and have no YAP1 expression, which is consistent with George et al. findings. The model we proposed is that in *Ngn3*-expressing cells, *Lats1/2* are required to keep YAP1/TAZ out of nuclei so that *Ngn3* can repress YAP1 expression. Loss of *Lats1/2* led to high nuclear YAP1 which may block *Ngn3*’s ability to suppress YAP1. ChIP-seq or ChIP-PCR on YAP1 promoter will answer the question whether YAP1 auto-regulates on the transcriptional level. However, the small number of cells in mouse pancreas limits us to perform this experiment.

– The numbers of animals used are unclear except for those depicted by the barplots. There is also no mention of the number of cells or fields analyzed for immunofluorescence experiments, which is essential for any quantitative comparisons and claims. The n's should be added throughout.

We added n’s to all figures and in our manuscript.

– The authors repeatedly refer to the dataset from Cebola et al. (Nat Cell Biol 2015), which was generated from human embryonic pancreatic progenitor cells, to speculate that TEAD1 binds to the promoter regions of YAP1, CDH1 and KRT19 and therefore may promote YAP1 autoregulation or expression of CDH1 and KRT19, explaining some of their immunofluorescence observations. However, they fail to acknowledge potential differences that may ensue due to the different species examined. Co-staining of Yap1 and Cdh1/Krt19 would indicate whether co-expression of Yap1 and Cdh1/Krt19 is indeed evident in the context of their study and provide further evidence to support their speculations.

Reviewer is correct that the dataset of TEAD1 ChIP-seq from Cebola et al. was generated from human embryonic pancreatic progenitor cells. These data are in line with our mouse experimental results where cells with high YAP1, due to loss of *Lats1&2,* continue to have high YAP1, CDH1 and KRT19. We do not have evidence to point out the potential differences between mouse and human.

– Several observations are over-interpreted or over-stated, and should be qualified as preliminary or speculative with proper wording. For example,P14: "differentiation…was blocked by lack of expression of ISL1 and NKX2.2". Although the authors observe low Isl1 and Nkx2.2 staining in NL vs control pancreases, no experiments were done to substantiate the claim that the reduction in ISL1 and NKX2.2 directly block the differentiation in this context.

We did not claim that the reduction in ISL1 and NKX2.2 directly block the differentiation in NL pancreas. ISL1 and NKX2.2 are markers for endocrine differentiation. The lack of ISL1 and NKX2.2 expression indicates that endocrine differentiation has been blocked.

– P15: "…KRT19 expression is not controlled by SOX9, but instead by YAP1". The authors observe that Krt19 proteins are increased in Lats1&2-null Ngn3+ cells, whereas Sox9 proteins were unchanged. However, they do not provide evidence that Yap1 controls Sox9 expression.

No *Sox9* expression was observed in *Ngn3*-expressing cells in both Control and NL pancreases (Figure 3B and 3C) while we observed YAP1 nuclei staining in NGN3-positive cells (Figure 5B), suggesting that YAP1 does not control *Sox9* expression.

Minor comments:– It is mentioned that deletion of Lats1&2 in Ngn3+ cells results in fewer acinar cells and smaller islets, evident by reduction in Ins+ or Gcg+ cells, whereas such genetic ablation in pancreatic β cells does not result in any phenotype. Did the deletion of Lats1&2 in Ngn3+ cells similarly lead to reduction in other endocrine cell types?

We showed that there was no positive staining for ISL1 and NKX2.2 in progeny of *Lats1&2* null *Ngn3*-expressing cells, suggesting the block of endocrine differentiation including all endocrine cells. The INS+ and GCG+ cells in NL pancreas are the escapers in which *Lats1&2* were not deleted. Other endocrine cells should be affected too. We observed smaller sized NL mice with low blood glucose levels. We have postulated that brain expression of *Ngn3^Cre^* may contribute to these phenotypes.

– The authors use the word "expression" without specification to refer to both mRNA expression and marker fluorescence levels throughout the text. This is inaccurate and potentially confusing. More specific terminologies should instead be used to avoid ambiguity.

We have added mRNA in the appropriate places where we discuss results from RT-qPCR experiments. All other places are protein expression results by immunostaining. We followed the general guideline for formatting gene and protein name throughout the manuscript: mouse gene symbols are italicized, with only the first letter in upper-case; protein symbols are not italicized, and all letters are in upper-case.

– Figure S1D is mis-referred to as S1E in the text.– Figure 7G is missing.– Table S1 is mis-referred to as S Table 2 in the text.– Typo on page 19: "Controlcontrol"– Figure S6B is mis-referred to as Figure S6A in the text.

We have made all appropriate changes to the manuscript to correct these mistakes.

[Editors’ note: what follows is the authors’ response to the second round of review.]

Based on the previous reviews and the revisions, the manuscript has been improved but there are some issues that need to be addressed, as outlined below:

The phenotype is clear although the authors don't dig deeper into the mechanism:

1. Is indeed YAP directly regulating Ngn3 expression or is Ngn3 directly regulating hippo? To my knowledge no one has done this. Maybe ChIP seq or at least ChIP-PCR will answer this.

Thanks for the suggestion. Ngn3 expressing is a transit state. The highest number of Ngn3 cells is at E15.5 mouse embryo but they are still a rare population in pancreas. It will be technical challenge to have enough cells for ChIP-seq or ChIP-PCR. The anti-Ngn3 antibodies have not been proved to be able to use for ChIP assay in mouse pancreas.

2. The phenotype is due to lack of delamination of endocrine cells from the trunk? inhibition of EMT?

Although the authors look at CDH1 expression, they don't look anything other than IF and include statements (without experiments) such as:

"These data suggest that an early effect of Lats1&2 deletion in NGN3+ cells is to activate KRT19 expression, but not *SOX9* expression, further indicating that KRT19 expression is not controlled by *SOX9*, but instead by YAP1"

(This is just based on *Sox9* and Krt19 IF…)

Thank reviewer’s question. We have shown in original manuscript that the Lats1&2 KO cells have high expression of KRT19 but not SOX9 (Figure 3B-C). In NL pancreas, CDH1 remains the high level of expression in Ngn3 expressing endocrine progenitor cells similar to ductal cells, suggesting that EMT or delamination is partially blocked. Our snRNA-seq has also shown that Cdh1 and Krt19 are highly expressed in mutant populations (Figure 6J-K).

3. Is Lats KO in Ngn3 cells forcing the bipotent trunk progenitor to exclusively differentiate into ducts?

To answer this question, we attempted to perform single cell RNA-seq analysis at our lab without success due to technical issues. We recruited Dr. Arda who is an expert on single cell analysis of pancreas. She used single nuclei RNA-seq (snRNA-seq) analysis and overcome technical issues. Based on the analysis, we found that Lats1/2 KO in Ngn3 cells forced the bipotent trunk progenitor to differentiate into a new population with ductal markers such as Onecut2 and Krt19 (new Figure 6J-K) but have their unique gene expression which was highly enriched for pathways including tissue migration, wound healing and Erk cascade (Figure S6E-F).

– Why is acinar differentiation affected if they floxed Lats by using Ngn3Cre? Is a paracrine effect? Cell to cell communication?

Our snRNA-seq analysis identified two acinar populations in the normal control pancreas: one mature population with high levels of enzyme gene expression and another less mature with lower levels of enzyme gene expression. In the NL pancreas, we observed a significant reduction in the mature population, accompanied by the emergence of a new population resembling the immature one, also characterized by lower enzyme gene expression. We hypothesize that the abundance of immune cells in the NL pancreas may lead mature cells to downregulate enzyme gene expression, mirroring the response seen in adult acinar cells during inflammation.

– No proof of Lats inactivation is showed in Ngn3Cre mice

PCR analysis on P1 NL pancreas confirmed deletions of both Lats 1 and 2. Additionally, Hippo signaling pathway enrichment was observed in the newly identified Ductal 4 population within the NL pancreas, as detailed in Figure 6J-K and Figures S6E-G.

– The authors talk about PSC activation and paracrine communication but don't check anything to indeed validate his point.

We observed expanded ACTA2 positive mesenchymal cells in NL pancreas (Figure S4A-B). We also observed the unique myofibroblast population in NL pancreas enriched genes for extracellular matrix organization, collagen formation and ECM-receptor interaction (Figure S6E).

– In general, the analysis can be extended. Unfortunately, the authors base all the statements by looking at stainings (not even RNAseq to look at DE genes).

We made the effort to perform snRNA-seq of our control and NL pancreas. The new experiments generated data as new Figure 6 and Figure S6. We added the result section, method section and a new paragraph of discussion. We believe these data confirmed our histology and staining data.